# Gametic specialization of centromeric histone paralogs in *Drosophila virilis*

Lisa E Kursel[1,2], Hannah McConnell[2], Aida Flor A de la Cruz[2,3], Harmit S Malik[2,3]

In most eukaryotes, centromeric histone (CenH3) proteins mediate mitosis and meiosis and ensure epigenetic inheritance of centromere identity. We hypothesized that disparate chromatin environments in soma versus germline might impose divergent functional requirements on single CenH3 genes, which could be ameliorated by gene duplications and subsequent specialization. Here, we analyzed the cytological localization of two recently identified CenH3 paralogs, Cid1 and Cid5, in *Drosophila virilis* using specific antibodies and epitope-tagged transgenic strains. We find that only ancestral Cid1 is present in somatic cells, whereas both Cid1 and Cid5 are expressed in testes and ovaries. However, Cid1 is lost in male meiosis but retained throughout oogenesis, whereas Cid5 is lost during female meiosis but retained in mature sperm. Following fertilization, only Cid1 is detectable in the early embryo, suggesting that maternally deposited Cid1 is rapidly loaded onto paternal centromeres during the protamine-to-histone transition. Our studies reveal mutually exclusive gametic specialization of divergent CenH3 paralogs. Duplication and divergence might allow essential centromeric genes to resolve an intralocus conflict between maternal and paternal centromeric requirements in many animal species.

## Introduction

Chromosome segregation is an essential process that is highly conserved across eukaryotes. Condensed chromosomes attach to the spindle via a specialized region of chromatin called the centromere, ensuring equal partitioning of DNA into daughter cells. Centromeres are defined by the centromeric histone variant, CenH3, which is the foundational centromeric protein in most eukaryotes (Sullivan et al, 1994; Yoda et al, 2000; Schuh et al, 2007b). First identified as Cenp-A in mammals (Earnshaw & Rothfield, 1985; Palmer et al, 1991), CenH3 localizes to centromeric DNA and helps recruit other components of the kinetochore, which mediates

chromosome segregation. The loss of CenH3 results in catastrophic chromosome segregation defects and lethality in protists, yeast, flies, nematodes, mice, and plants (Stoler et al, 1995; Buchwitz et al, 1999; Howman et al, 2000; Blower & Karpen, 2001). Although some lineages lack CenH3 altogether (Akiyoshi & Gull, 2014; Drinnenberg et al, 2014), in most eukaryotes that encode CenH3, it is essential for chromosome segregation in both mitosis and meiosis.

In addition to CenH3's critical role in mitotic and meiotic chromosome segregation, CenH3 protein retention is important for the epigenetic inheritance of centromere identity through spermiogenesis. During the production of male gametes in many animal species, the sperm nucleus undergoes a dramatic transition from histone-based chromatin to chromatin that is packaged by protamines; nearly all of the histones are removed and are replaced by highly basic proteins called protamines (Oliva & Dixon, 1991; Braun, 2001; Renkawitz-Pohl et al, 2005). Even though CenH3 is a histone protein, it is not removed from sperm chromatin during this process. Studies in mammals find the presence of CenH3 in mature sperm (Palmer et al, 1990). Furthermore, loss of paternal CenH3 on sperm chromatin in *Drosophila melanogaster* results in early embryonic lethality (Raychaudhuri et al, 2012). Thus, CenH3 needs to function in disparate chromatin environments in multicellular animals, in a histone-rich environment in somatic cells and in a protamine-rich environment in sperm, which may impose divergent functional constraints on CenH3.

The female germline could also impose distinct constraints on CenH3 function, particularly in long-lived animals. In humans and mice, oocyte nuclei arrest in meiotic prophase I for extended periods of time (years in humans, months in mice) (Von Stetina & Orr-Weaver, 2011; Smoak et al, 2016). Oocyte centromere function does not seem to depend on the loading of newly transcribed CenH3 as conditional knockouts of CenH3 in meiotic prophase I are fully fertile in *Mus musculus* (Smoak et al, 2016). However, recent work demonstrated that CenH3 in Meiosis I (MI) arrested starfish oocytes undergoes gradual turnover, presumably to replace CenH3 containing nucleosomes that are disturbed by transcriptional machinery, allowing oocytes to maintain centromere competence over long periods of time (Swartz et al, 2019). This means that CenH3

[1]Molecular and Cellular Biology Graduate Program, University of Washington, Seattle, WA, USA   [2]Division of Basic Sciences, Fred Hutchinson Cancer Research Center, Seattle, WA, USA   [3]Howard Hughes Medical Institute, Fred Hutchinson Cancer Research Center, Seattle, WA, USA

Correspondence: hsmalik@fhcrc.org
Lisa E Kursel's present address is Ofer Rog Lab, Department of Biology, University of Utah, Salt Lake City, UT, USA
Hannah McConnell's present address is Department of Biology, University of Washington, Seattle, WA, USA

molecules are capable of stably persisting in oocytes for long periods of time and that there are mechanisms in place to maintain centromere function in nondividing cells.

These separate functional requirements could impose opposite selective constraints on CenH3. For instance, one might anticipate that CenH3's essential mitotic function would lead to functional constraint and strong amino acid conservation ("purifying selection"). Contrary to this expectation, CenH3 has been found to evolve rapidly in many species of plants and animals (Malik & Henikoff, 2001; Talbert et al, 2004; Schueler et al, 2010). We previously hypothesized that this rapid evolution is a result of CenH3's role as a suppressor of centromere drive. Centromere drive results from an inherently asymmetric transmission of chromosomes through female meiosis in both plants and animals (Malik, 2009; Kursel & Malik, 2018). In the first step of this process, chromosomes "selfishly" compete via centromeric protein recruitment to bias meiotic spindle orientation to preferentially transmit themselves into the egg rather than to polar bodies. In the second step of the model, centromeric proteins adaptively evolve to restore meiotic parity between competing chromosomes and suppress the deleterious effects of centromere drive (Henikoff & Malik, 2002; Malik, 2009; Kursel & Malik, 2018).

Because of these disparate functions, CenH3 proteins may have different protein-coding requirements in different cellular contexts, especially in the germline (Das et al, 2017; Prosee et al, 2020). Dissecting these multiple functional constraints in many model organisms (such as *D. melanogaster* and *M. musculus*) is challenging because CenH3 is an essential single-copy gene in these species. However, in organisms in which CenH3 has duplicated, these specialized functions may be partitioned among paralogs. Thus, gene duplications present a unique opportunity to more precisely understand the tissue-specific functions of CenH3. Indeed, some plant species have multiple CenH3 paralogs (Kawabe et al, 2006; Finseth et al, 2015; Ishii et al, 2015; Maheshwari et al, 2015) that often show signs of tissue-specific specialization. For example, knockdown of one CenH3 paralog in wheat causes growth defects whereas knockdown of the other paralog causes reproductive defects (Yuan et al, 2015). However, the molecular basis of this specialization is unclear.

In contrast to plants, centromeric histone specialization has not been previously observed in animal species. Although an estimated 10% of plant genomes harbor multiple CenH3 paralogs (Kawabe et al, 2006; Finseth et al, 2015; Maheshwari et al, 2015), CenH3 duplications were previously thought to be rare in animals (Li & Huang, 2008; Monen et al, 2015). Contrary to this view, the CenH3 gene (known as *Cid*) has duplicated at least four times in *Drosophila* (Kursel & Malik, 2017) and at least three times in mosquitoes (Kursel et al, 2020). All mosquito species and most *Drosophila* species encode more than one *Cid* gene. *Drosophila* and mosquito *Cid* paralogs can localize to centromeres when ectopically expressed, but many paralogs have evolved germline-restricted expression patterns, highly divergent N-terminal tails, and divergent selective constraints. This discovery led us to hypothesize that Cid paralogs have independently acquired tissue or cell type–specific functions in both *Drosophila* and mosquito species (Kursel & Malik, 2017; Kursel et al, 2020).

To test this hypothesis, we performed cytological analysis of the two *Cid* paralogs in *Drosophila virilis*, *Cid1* and *Cid5*, which diverged nearly 40 million years ago and have since been co-retained in the *Drosophila* subgenus (Kursel & Malik, 2017). We examined Cid1 and

Cid5 localization in *D. virilis* somatic cells, testes, ovaries and early embryos. We found that there is mutually exclusive retention of the two Cid proteins in mature male and female gametes, which is achieved by protein loss of different paralogs during meiosis in males and females. We confirm that *D. virilis* sperm only carry Cid5 protein on their centromeres, which disappears before the first mitotic division in the zygote. We hypothesize that paralog-specific changes in the N-terminal domain have allowed for the functional specialization of Cid1 and Cid5. Thus, Cid paralogs in *D. virilis* appear to have used gene duplication and specialization to resolve the tension of multiple, disparate CenH3 functions. This specialization further suggests that single copy CenH3 proteins may not represent the most optimal state in multicellular, sexual organisms.

# Results

The ancient retention of *Cid1* and *Cid5* suggests that both paralogs perform important, nonredundant, functions in *D. virilis* and related species (Kursel & Malik, 2017). To gain insight into the function of Cid1 and Cid5, we investigated their localization in dividing somatic cells, ovaries, testes, and embryos of *D. virilis* flies. For this approach, we developed tools to visualize Cid1 and Cid5 in vivo. We exploited the high divergence of their N-terminal tails to develop polyclonal antibodies that are specific to either Cid1 or Cid5 (Fig S1A). We confirmed that each antibody specifically recognized the paralog it was designed for via immunofluorescence analyses (Fig S1B).

Because antibody occlusion could hamper cytological analyses especially in the male germline (Bonnefoy et al, 2007), in parallel we also generated transgenic *D. virilis* flies with *Cid1GFP* or *Cid5mCherry* under the control of their respective native promoters. In *D. melanogaster*, *Cid-GFP* transgenic flies, in which GFP was inserted between the N-terminal tail and histone fold domain (HFD) of *Cid*, can complement *Cid* function (Schuh et al, 2007a). Therefore, we inserted the fluorescent protein tag between the N-terminal tail and the HFD in both Cid1 and Cid5 transgenes (Fig S1C).

## Cid1, but not Cid5, is detectable in somatic cells

Our previous expression analyses based on qRT-PCR (Kursel & Malik, 2017) found that *Cid1* is expressed in somatic cells including *D. virilis* WR-Dv-1 tissue culture cells (derived from first instar larvae), heads from male and female *D. virilis* flies and male and female carcasses (excluding heads and gonads), whereas *Cid5* is not. To examine protein expression, we looked for Cid1 and Cid5 protein in two types of dividing somatic cells: tissue culture cells and larval neuroblasts. In WR-Dv-1 tissue culture cells, we could detect endogenous Cid1 protein by both Western blot and immunofluorescence analyses (Fig 1A and B). However, we did not detect Cid5 using either method (Fig 1A and C), consistent with our previous finding that *Cid5* RNA is not found in these cells (Kursel & Malik, 2017).

Next, we examined Cid1 and Cid5 localization in larval neuroblasts, a tissue that is enriched in mitotic cells. As expected, we found that Cid1 localized to centromeres in interphase cells and on

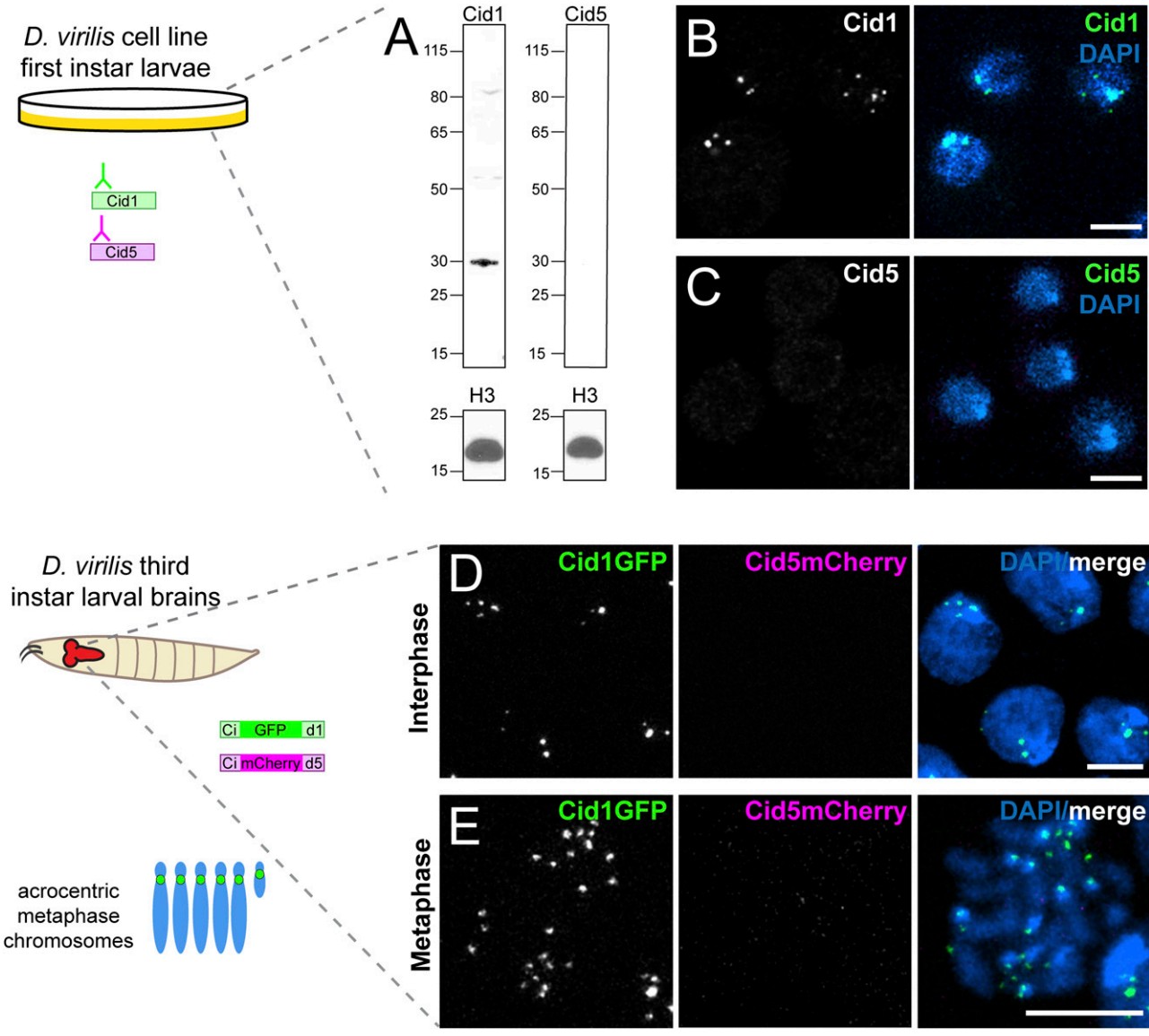

**Figure 1.  Cid1 is the centromeric histone in two dividing somatic cell types.**
**(A)** Western blot of Cid1 and Cid5 in *Drosophila virilis* tissue culture cells. A Western blot for histone H3 was used as a loading control. This Western blot was repeated three times with the same result (Fig 1 source file). Uncropped, original color scans of Western blots can be found in source data. **(B, C)** Immunofluorescence images of *D. virilis* tissue culture cells stained with Cid1 (B) or Cid5 (C) antibodies. **(D, E)** Images of interphase (D) or metaphase (E) nuclei from *D. virilis* larval brains dissected from flies containing Cid1GFP and Cid5mCherry transgenes. Scale bar = 5 μm.
Source data are available for this figure.

condensed metaphase chromosomes (Fig 1D and E). As *D. virilis* chromosomes are acrocentric (have their centromeres close to one telomere), the Cid1 signal was localized to one end of each condensed chromosome. In contrast, we could not detect any Cid5 signal (Fig 1D and E). Our cytological findings using transgenes were confirmed by detection using polyclonal antibodies, reinforcing the validity of our transgene analyses (Fig S1B).

## Differential retention of Cid1 and Cid5 in *D. virilis* ovaries

We next investigated Cid1 and Cid5 protein localization in *D. virilis* ovaries. The *Drosophila* ovary is made up of about 16 ovarioles. At

the anterior tip of each ovariole, germline stem cells (GSCs) divide four times to produce a cyst of 16 interconnected cells, which differentiate into 15 nurse cells (support cells that provide mRNA, protein, and other material to the oocyte via a shared cytoplasm) and one oocyte. These interconnected germ cells are surrounded by somatic follicle cells and together form an egg chamber. Egg chamber maturation occurs progressively along the ovariole in a series of defined stages. These stages are referred to as stages 1–14 based on growth and organization of somatic and germline cells. By stage 2, the oocyte has entered into meiotic prophase and reaches pachytene. At stage 5, the oocyte enters primary arrest and remains arrested until stage 13 when the oocyte progresses to secondary

arrest in metaphase of meiosis I. In the stage 14 egg chamber, no nurse cell nuclei remain and the oocyte is prepared for ovulation (King et al, 1956; Spradling, 1993). Our previous study showed that *Cid1* transcripts were abundant but *Cid5* transcripts were not detectable in RNA extracted from whole ovaries (Kursel & Malik, 2017). We, therefore, expected to find that Cid1 would be the only Cid paralog detectable in *D. virilis* ovaries.

To examine Cid1 and Cid5 protein, we used Cid1GFP and Cid5mCherry transgenic flies and Cid1 and Cid5 antibodies for localization of both proteins in somatic and germline cells at different stages of egg chamber development. Similar to mitotically dividing somatic cells, we detected Cid1 but not Cid5 in somatic terminal filament and follicular cells (Figs 2A and B and S2A). However, we unexpectedly detected both Cid1 and Cid5 protein in the germline lineage cells of the ovary including in GSCs (Figs 2C and S2B), nurse cells (Figs 2D and S2A), and the oocyte nucleus (Figs 2E and S2A).

Cid1GFP remained on the oocyte nucleus through stage 14 (MI – metaphase) oocytes but we could not reliably detect Cid5mCherry at this stage (Fig 2F). We examined this loss of Cid5 in more detail and found that we could detect Cid5mCherry in mid- to late-stage oocytes (stage 8, 10, 12, Fig S2C–E), but only in two out of six stage 14 oocytes (Fig S2F and G). We substaged the stage 14 nuclei based on the distance across the long-axis of the oocyte nucleus under the assumption that a later stage 14 nucleus will be maximally stretched on the metaphase plate. We found that the two nuclei that contained Cid5mCherry were the earliest stage 14 oocytes whereas later stage 14 oocytes lacked detectable Cid5mCherry (Fig S2G). This suggests that Cid5 is rapidly removed as the chromosomes are pulled toward opposite poles in MI—metaphase. We similarly detected Cid1 and Cid5 protein in germline cells using the Cid1 and Cid5 antibodies (Fig S2C and H). However, antibody staining against either Cid protein was not successful in stage 14 egg

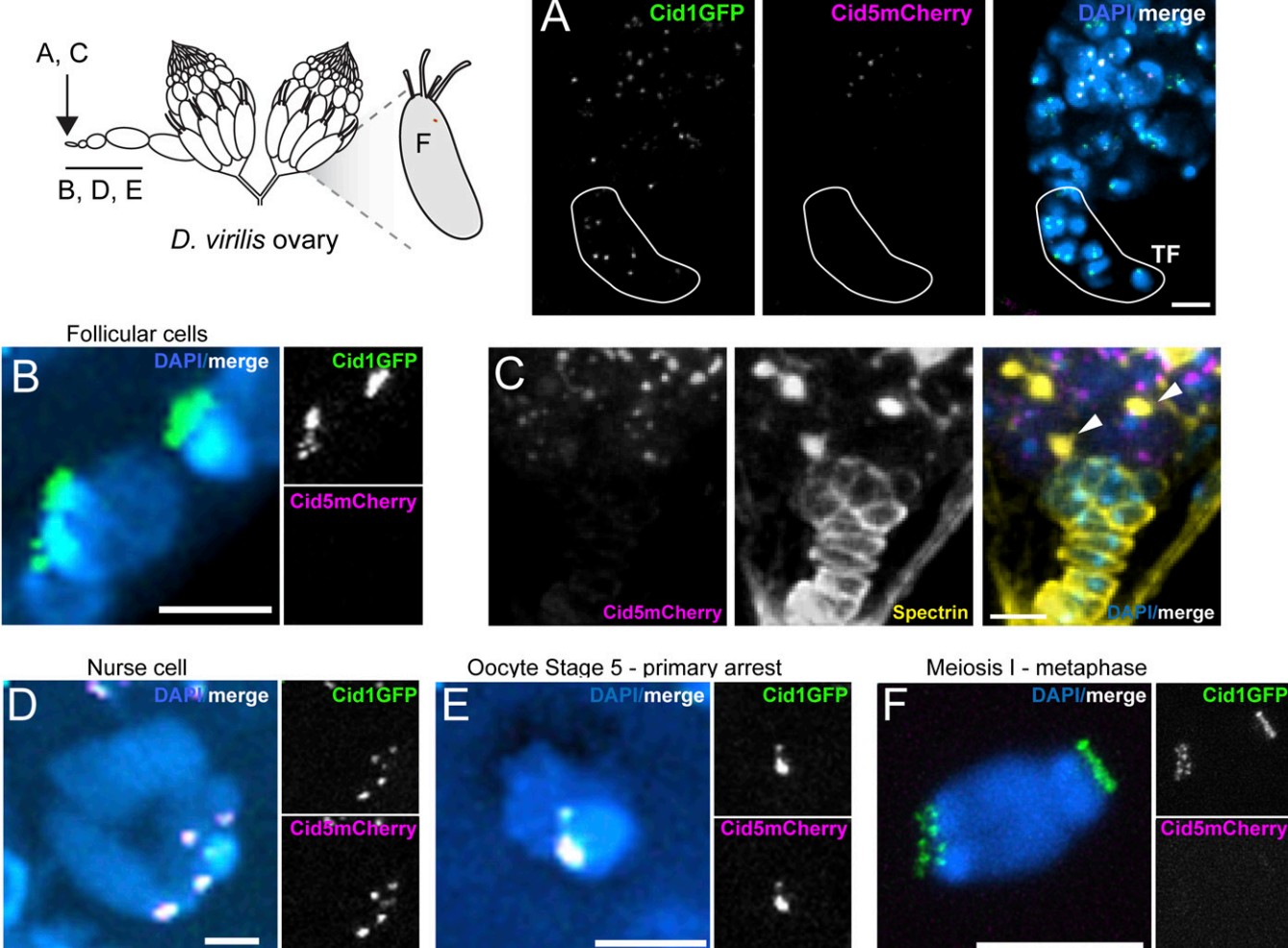

**Figure 2. Differential localization of Cid1 and Cid5 in ovaries.**
All images use Cid1GFP and Cid5mCherry to detect Cid1 and Cid5 protein. **(A)** Image of the apical tip of the germarium showing Cid1GFP and Cid5mCherry localization. Somatic terminal filament cells (TF) are circled with a solid white line. **(B)** Image of somatic follicular cells from a stage 3 egg chamber. **(C)** Image of apical tip of the germarium stained with anti-Spectrin. White arrowheads point to the spectrosomes in the merge panel. **(D)** Image of a nurse cell from a stage 4 egg chamber. **(E)** Image of a stage 5 oocyte in primary arrest. **(F)** Image of a stage 14 oocyte nucleus in the MI—metaphase arrest. Uncropped images of panels (B, D, E) can be found in Fig S2A. Uncropped image of panel (C) can be found in Fig S2B. All scale bars = 5 µm.

chambers (Fig S2I) likely because of antibody accessibility, high-lighting the utility of our dual cytological approaches.

Our results suggest that both Cid1 and Cid5 are present at centromeres early in oogenesis. Given that turnover of CenH3-containing nucleosomes in MI-arrested oocytes appears to be quite gradual (~2% of centromeric CenH3 is exchanged per day in MI-arrested starfish oocytes [Swartz et al, 2019]), we hypothesize that Cid5 protein is actively removed from the oocyte centromeres at the onset of meiosis I metaphase arrest. Because Cid1 is always present throughout oogenesis, it is unclear whether Cid5 performs any nonredundant function, centromeric or otherwise, in the female germline. However, it is apparent that Cid1 is the only detectable centromeric histone in late-stage *D. virilis* oocytes and is therefore likely to be essential for female fertility and early embryonic mitotic divisions after fertilization.

### Differential retention of Cid1 and Cid5 in *D. virilis* testes

Our previous characterization of *Cid1* and *Cid5* mRNA expression in *D. virilis* (Kursel & Malik, 2017) indicated that both *Cid* paralogs are expressed in testes. We therefore examined the cytological localization patterns of Cid1 and Cid5 in the *D. virilis* male germline. We also used GFP-Hiphop as a telomeric marker (Gao et al, 2011). Since *D. virilis* flies have acrocentric chromosomes, their centromeric cytological signals should be adjacent to one of the two telomeric, GFP-Hiphop–labeled, cytological signals on each chromosome. Thus, Hiphop localization serves as an additional centromere-adjacent marker in *D. virilis*.

In the *Drosophila* male germline, spermatogenesis begins at the apical tip of the testis where the GSCs reside. The asymmetric divisions of the GSCs replenish the stem cell population and produce gonialblasts. These gonialblasts divide mitotically with incomplete cytokinesis and then enter an extended meiotic prophase. After this extended period of cell growth, cysts of 16 spermatocytes undergo meiosis and produce bundles of 64 haploid spermatids (Fuller, 1993; Fabian & Brill, 2012). These spermatids then go through the process of nuclear remodeling resulting in 200-fold compaction of their nuclear volume (Fuller, 1993). During this dramatic nuclear reorganization, nearly all of the histones are removed and are replaced by sperm nuclear basic proteins (Renkawitz-Pohl et al, 2005). Finally, elongated spermatid bundles go through individualization to produce mature sperm (Fig 3A and B).

Previous studies in *D. melanogaster* have shown that Cid is essential for the mitotic and meiotic divisions in the male germline (Dunleavy et al, 2012). Moreover, Cid has also been shown to be critical for transgenerational centromere inheritance through the mature sperm (Raychaudhuri et al, 2012). Therefore, we examined the cytological localization of Cid1 and Cid5 in the mitotic zone, meiotic zone, post-meiotic stages, and in mature sperm (Fig 3A and B). We examined testes from Cid5mCherry males and performed antibody staining with the Cid1 antibody, a spectrin antibody to identify the GSCs and a phospho-histone H3 Serine 10 (PH3S10) antibody to identify condensed chromosomes (Hendzel et al, 1997; Tang et al, 1998; Ivanovska & Orr-Weaver, 2006). We found that Cid1 and Cid5 co-localize at centromeres in most mitotic zone nuclei (Figs 3C and S3A). However, we noticed that Cid1 was observable in a

group of cells close to the apical tip of the testis whereas Cid5 was not (Fig 3C, inset). We hypothesized that these Cid5-negative cells make up the somatic "hub" cells (Kiger et al, 2001). Indeed, we found that Cid5mCherry was absent from the floret shaped hub but present in neighboring GSCs (Fig 3D). This suggests that Cid5 first appears in the GSCs and its localization is restricted to germ-lineage cells in the testis like in the ovary.

We followed the localization of Cid1 and Cid5 through meiotic prophase where they co-localize at centromeres in primary spermatocytes. Surprisingly, at the onset of metaphase of meiosis I, Cid1 was no longer detectable; we could only detect Cid5 on these chromosomes (Fig 3E). The disappearance of Cid1 seemed to be rapid and coincide with the appearance of PH3S10 on condensing chromosomes (Fig 3F). We found that Cid5 persists in post-meiotic stages as a discrete focus on each "leaf-stage" and "late-canoe-stage" spermatid nucleus (Fabian & Brill, 2012), but we never observed Cid1 at these stages (Figs 3G and H and S3B). To confirm that our inability to detect Cid1 on condensing meiotic chromosomes and post-meiotic spermatids was not due to antibody accessibility issues, we also examined Cid1GFP in the male germline. The results were nearly identical to the antibody staining. We robustly detected Cid1 at centromeric foci in the mitotic zone (Fig S3C) but only faintly in cells entering metaphase of meiosis I zone (Fig S3D). We could not detect Cid1 at any stage after meiosis, including in mature sperm (Fig S3E–G). Our results are thus consistent between our antibody staining and transgene analyses, except for cells entering meiosis I metaphase, in which Cid1GFP is either slightly more sensitive than the Cid1 antibody or persists longer than endogenous Cid1.

Our findings indicate that metaphase of meiosis I represents a transition state between the presence of Cid1 in mitotic and early meiotic cells and its absence in post-meiotic cells. Like the loss of Cid5 in oocytes, this loss of Cid1 occurs without DNA replication, suggesting an active protein degradation mechanism may be responsible. Interestingly, previous studies in *D. melanogaster* testes also observed a decrease in Cid levels coinciding with changes in kinetochore organization and orientation between meiosis I and meiosis II (Dunleavy et al, 2012). Thus, metaphase of meiosis I represents a centromeric transition state in both males and females, except that Cid5 is specifically lost in the female germline and Cid1 is specifically lost in the male germline.

Our cytological analyses further indicate that Cid5's centromeric localization persists throughout male gametogenesis from early germ cells to sperm. Previous findings have demonstrated that Cid protein is required for transgenerational inheritance of centromere identity through sperm in *D. melanogaster* (Raychaudhuri et al, 2012). Because Cid1 is not detectable during spermiogenesis, we hypothesized that Cid5 might provide the transgenerational centromeric mark in mature sperm in *D. virilis*. To further investigate Cid5 localization in *D. virilis* sperm and validate its centromeric localization, we examined the localization of GFP-HipHop and Cid5mCherry in the testes of flies that contained both transgenes. We observed two primary HipHop foci corresponding to telomeric ends in each spermatid nucleus (Fig 4A and B). We also saw a single Cid5 focus, which consistently co-localized with one of the two GFP-HipHop foci (Fig 4C). This localization pattern persisted throughout spermatid development and in mature sperm (Fig 4C–E). These

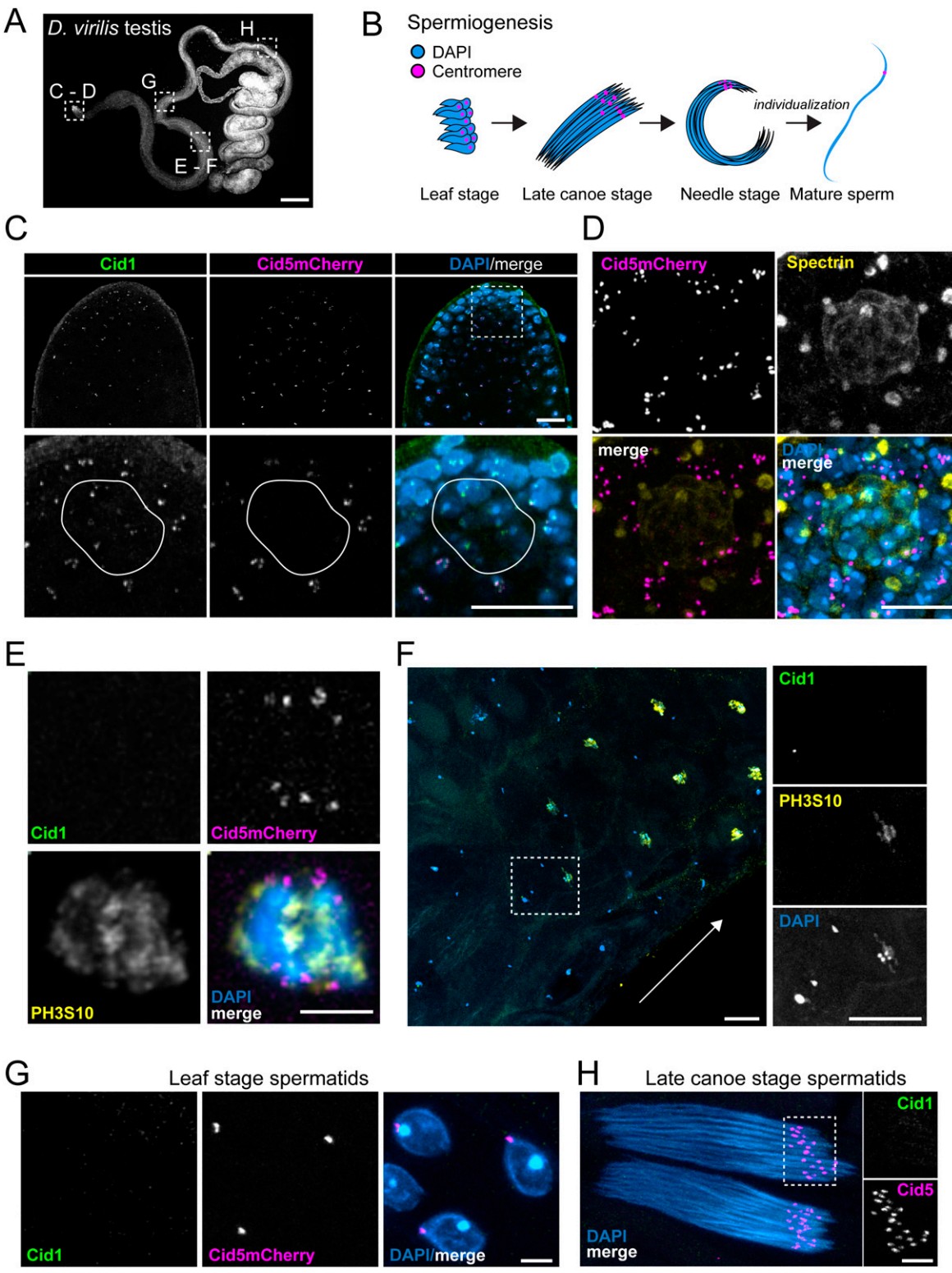

**Figure 3. Differential localization of Cid1 and Cid5 in testes.**
**(A)** Image of a DAPI-stained *Drosophila virilis* testis. **(C, D, E, F, G, H)** Boxed regions show the approximate location of panels (C, D, E, D, F, G, H). Scale bar = 100 μm. **(B)** Schematic showing stages of spermiogenesis. **(C)** The apical tip (mitotic zone) of a *D. virilis* testis. The bottom panel shows a high magnification image of the area indicated in the top panel by the dashed box. The solid white line outlines nuclei that contain Cid1 but not Cid5. Scale bar = 25 μm. **(D)** The germline stem cell (GSC) niche stained with anti-Spectrin to identify the hub cells (rosette structure) and the GSCs. GSCs are next to the hub cells and contain a large spectrin-positive structure called the spectrosome. Cid5mCherry signal is absent from the hub cells but present in the surrounding GSCs. Scale bar = 10 μm. **(E)** A single cell with condensing chromosomes in

experiments give additional support to the hypothesis that Cid5 provides the transgenerational centromere mark in *D. virilis*—Cid5 is present at centromeres in mature sperm, but Cid1 is not.

Taken together, our cytological examination of Cid1 and Cid5 suggests that after prometaphase, Cid5 is the predominant centromeric histone in the *D. virilis* male germline. Our inability to detect Cid1 in post-prometaphase meiotic cells and post-meiotic spermatids suggests that male meiotic and centromere inheritance function in *D. virilis* flies does not require Cid1, even though the *D. melanogaster Cid1* ortholog, *Cid*, is essential for both processes (Dunleavy et al, 2012; Raychaudhuri et al, 2012). Thus, male and female gametes alternately retain different Cid protein paralogs in *D. virilis*.

### Cid1 is the only detectable centromeric histone in the early embryo

Our cytological analyses revealed that the mature oocyte nucleus in *D. virilis* only retains Cid1 whereas mature sperm only retain Cid5 (Fig 5A). We next investigated how parental genomes with distinct Cid paralogs coordinate chromosomal events in the early embryo. One of the most dramatic chromosomal changes following fertilization is the remodeling of the sperm nucleus, in which protamines that package the bulk of sperm chromatin are replaced with maternally provided core and variant histones in a replication-independent manner (Loppin et al, 2000, 2001, 2005). While paternal chromosome remodeling occurs, female meiosis is completed. Maternal and paternal pronuclei then congress towards each other, appose, and undergo mitosis synchronously but on separate halves of the first spindle (Fig 5A). Defects in this synchronization lead to embryonic lethality (Landmann et al, 2009; Levine et al, 2015).

In *D. melanogaster*, paternal Cid persists on the paternal genome throughout this extensive remodeling and is required for the first embryonic cell divisions, even though the specific molecules of paternal Cid only persist until the third embryonic cell cycle (Raychaudhuri et al, 2012). However, Cid1 is completely removed from paternal genomes during spermatogenesis in *D. virilis*, whereas Cid5 remains. Despite this paternal inheritance, we never observed Cid5 in any somatic cells, implying that it must either be removed or significantly diluted during zygotic development. Therefore, we examined the dynamics of sperm-inherited Cid5 and maternally deposited Cid1 on the parental genomes after fertilization in *D. virilis*.

Based on the precedent of CidGFP retention in *D. melanogaster*, we expected that paternally inherited Cid5 would persist on the paternal genome through the first several embryonic cell cycles, whereas Cid1 would define centromeres throughout the completion of female meiosis, co-localize with Cid5 in the early embryo and gradually replace Cid5 to eventually become the only Cid protein present in the embryo. To test this hypothesis, we examined Cid1GFP and Cid5mCherry in embryos produced by male and female parents bearing both transgenes. Consistent with our previous findings that meiosis I metaphase arrested oocytes only contain Cid1 (Fig 2F), we found that Cid1 is detectable on the maternal genome through the completion of meiosis (Fig 5B and C). More surprisingly, we were only able to only detect Cid1 on the paternal pronucleus, even at very early stages (Fig 5B and C) despite our earlier observations that mature sperm only contain readily detectable Cid5 (Fig 4E). At the earliest embryonic stage imaged (MII – metaphase), the paternal genome already stained positive for Acetylated histone H4 (Fig S4A) indicating that the protamine to histone transition has already occurred. This suggests that Cid5 is removed from the paternal genome before or concurrent with the protamine to histone transition.

Although Cid1 signal was faint on the paternal genome at earlier stages, it reached a level comparable with the Cid1 signal on the maternal genome by the time of apposition and the synchronous first mitosis (Fig 5B–F, compare paternal and maternal Cid1 foci in Fig 5E and F), suggesting that Cid1 gradually builds up on the paternal genome before the first embryonic mitosis. Our data are also consistent with the loading of Cid during early embryonic anaphase in *D. melanogaster* (Schuh et al, 2007b). However, in contrast to *D. melanogaster*, our results suggest that the initial loading of maternal Cid1 onto paternal chromosomes occurs more rapidly in *D. virilis*, during the protamine-to-histone chromatin transition before the first mitotic division.

We validated maternal versus paternal deposition of specific Cid paralogs by crossing parents that each encoded only one of the two tagged Cid transgenes. We first crossed Cid1GFP-encoding females to Cid5mCherry-encoding males (Fig S4B and C). Consistent with our previous findings (Fig 5), we found that these early embryos only have Cid1GFP at their centromeres, either in the M phase (Fig S4B) or the in S phase (Fig S4C). Next, we performed reciprocal crosses, between Cid5mCherry-encoding females and Cid1GFP-encoding males. As expected, we observed no Cid1GFP or Cid5mcherry on any centromeres in resulting embryos (Fig S4D); Cid1GFP was presumably removed during male meiosis and Cid5mCherry was removed during female meiosis resulting in no fluorescent protein in the early embryo. Our findings confirm the rapid loading of Cid1 onto the paternal genome immediately after fertilization. They also validate our conclusion that no detectable Cid1 protein is inherited via *D. virilis* sperm. These results underscore the mutually exclusive functional specialization of the two *Cid* paralogs in *D. virilis*.

## Discussion

Our study suggests that *D. virilis* uses a dedicated CenH3 paralog, *Cid5*, specifically for the purpose of epigenetic inheritance of centromere identity through the protamine-rich sperm chromatin environment (Fig S5). This specialization was accomplished after

---

late prometaphase or early metaphase of MI. Scale bar = 5 $\mu$m. **(F)** Image of primary spermatocytes entering MI—metaphase. Arrow indicates direction of meiotic progression. Inset shows a primary spermatocyte immediately adjacent to condensing meiotic chromosomes (marked by PH3S10). Cid1 is observable on the primary spermatocyte but not on the condensing meiotic chromosomes. Scale bar = 10 $\mu$m. **(G)** Leaf-stage spermatid nuclei. **(H)** Late-canoe stage spermatid bundles. Scale bars = 5 $\mu$m in (G) and (H).

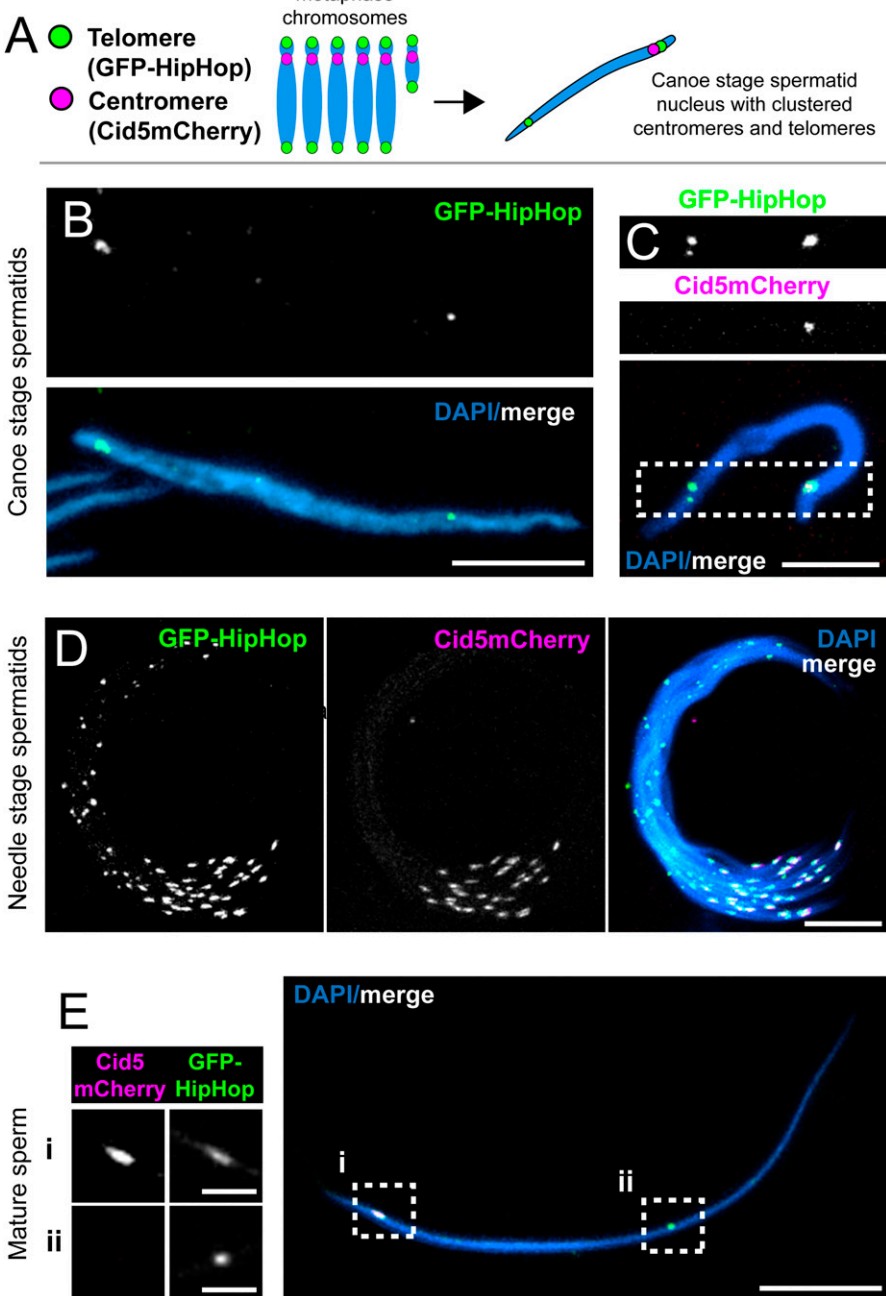

**Figure 4. Cid5 provides the centromere mark in mature sperm.**
**(A)** Schematic showing haploid chromosomes (left) which become condensed into the Rabl configuration (right) in *Drosophila virilis* sperm. **(B)** A single late canoe stage spermtid from a GFP-HipHop fly. All subsequent panels show images from flies with both GFP-HipHop and Cid5mCherry transgenes. **(C)** A single late-canoe stage spermatid. **(D)** Needle-stage spermatid bundle. **(E)** A single mature sperm nucleus. Boxed regions (i) and (ii) are also shown at slightly higher magnification and as separate channels (left). Scale bars in (B, C, D) = 10 μm. Scale bar = 10 μm in (E) uncropped panel and 2.5 μm in insets (E-i and E-ii).

gene duplication by gain of germline-specific expression of Cid5 and specific removal of the ancestral Cid1 during the prometaphase to metaphase transition of MI in the male germline (Fig S5). It is technically possible that a small amount of Cid1 remains in mature sperm beyond our cytological limit of detection via multiple methods. However, degradation of *D. melanogaster* Cid in the male germline below the limits of cytological detection results in early embryonic lethality (Raychaudhuri et al, 2012). Thus, undetectably low levels of Cid1 are unlikely to be functionally sufficient for epigenetic inheritance of centromere identity. Therefore, we hypothesize that Cid5 must provide the epigenetic mark of

centromeres across generations. Although Cid1 no longer appears to perform centromere inheritance function through the male germline in *D. virilis*, it still serves CenH3 function in both the soma and the female germline.

Based on these findings, we predict that *D. virilis* flies lacking Cid1 would be inviable, just like *Cid* knockdown in *D. melanogaster* (Blower & Karpen, 2001) whereas Cid5 knockouts would result in either male sterility or paternal effect lethality as observed when *D. melanogaster* Cid is specifically depleted in sperm (Dunleavy et al, 2012; Raychaudhuri et al, 2012). However, it is formally possible that Cid1 may perdure through spermatogenesis in the absence of Cid5.

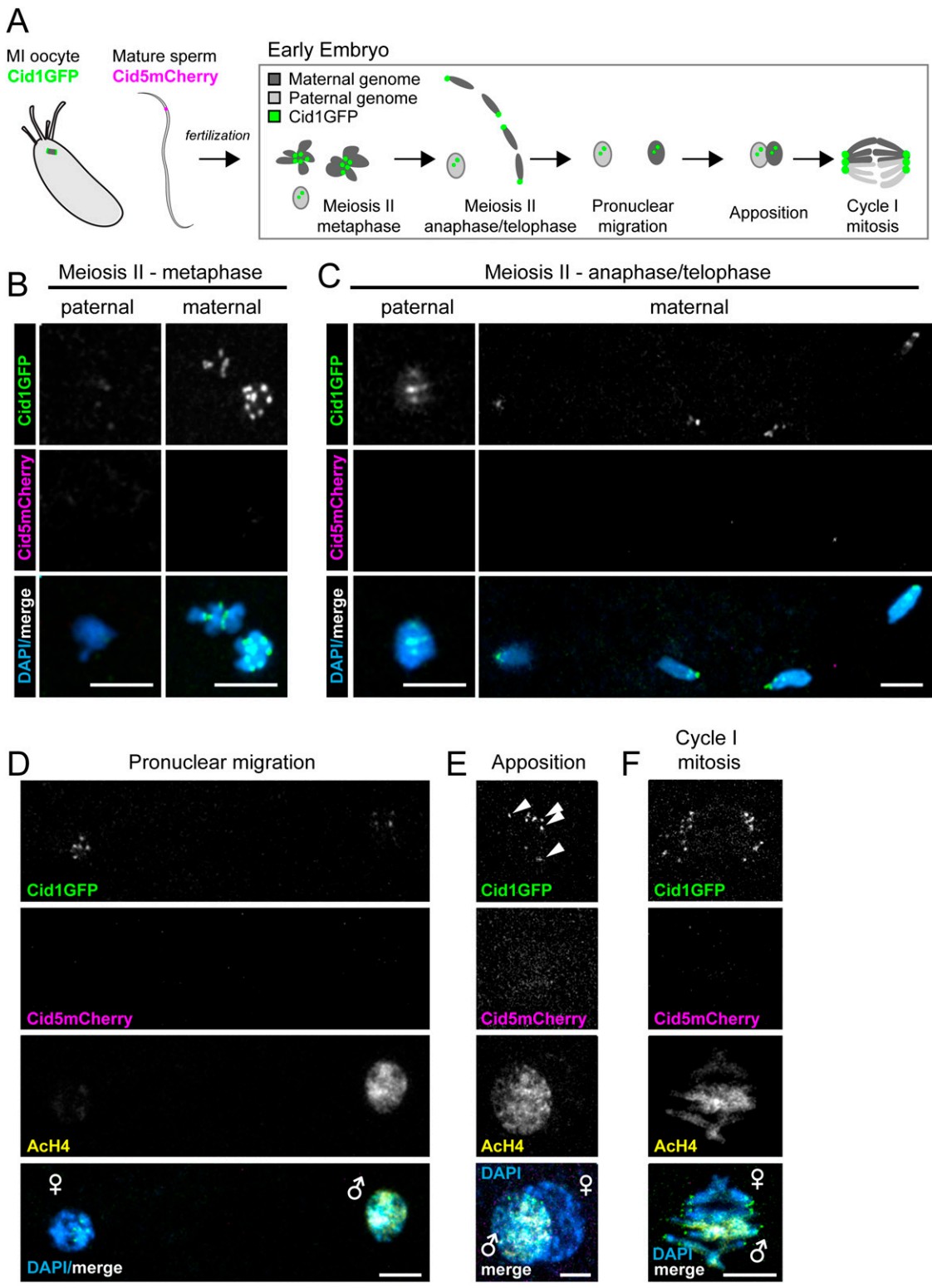

**Figure 5.  Cid1 is the primary centromeric histone in the early embryo.**
**(A)** Schematic of fertilization and the progression of the maternal and paternal genome in the early embryo through the first embryonic mitosis. All other panels are images from *Drosophila virilis* early embryos that were collected from parents that both had Cid1GFP and Cid5mCherry transgenes. **(B, C, D, E, F)** Paternal and maternal genomes were discerned by nuclear morphology, (B) and (C), or by acetylated histone H4 (AcH4) antibody staining, (D, E, F). AcH4 preferentially stains the paternal genome. **(B)** Meiosis II metaphase. **(C)** Meiosis II anaphase/telophase. **(D, E, F)** Pronuclear migration, apposition, and the first embryonic mitotic cell division. Arrowheads in (E) point to paternal Cid1GFP foci. All scale bars = 5 μm.

It is also possible that Cid1 is required for loading Cid5 onto centromeres in male germ cells based on its pattern of expression in GSCs and germ cells in the testis (Fig 3); in this case, Cid1 knockdown in the male germline would also result in male sterility. Unfortunately, despite several attempts, we have been unable to obtain reliable, robust knockdown or knockout of *Cid5* in *D. virilis* to directly test these predictions (the Materials and Methods section provides a detailed description of our efforts). Nevertheless, we anticipate that because of their specialization, knockout or genetic knockdown of *Cid1* and *Cid5* will have different phenotypic consequences, just as previously observed for wheat *CenH3* paralogs (Yuan et al, 2015).

Based on our earlier RT-PCR analyses, we expected to find no expression of Cid5 during oogenesis. However, we found strong evidence that Cid5 is expressed and localizes to centromeres in GSCs and germ cells early during oogenesis, only being lost at stage 14 of mature oocytes (Fig 2). It is currently unclear why Cid5 expression during oogenesis is important, if the protein does not survive oogenesis. One possibility is that this *Cid5* expression pattern is not functionally important but rather an evolutionary relic of its gene duplication from *Cid1* and subsequent specialization for germline-specific expression in both testes and ovaries. Thus, *Cid5*'s expression may have been driven by a GSC and germ cell–specific promoter that cannot distinguish between male and female germlines. However, evolutionarily young genes tend to be heavily biased towards testis-specific expression after birth, subsequently expanding their expression to other tissues (Assis & Bachtrog, 2013). If *Cid5* initially arose as a testis-specific gene that subsequently began to be expressed during oogenesis, it is possible that Cid5 currently performs some important function in nurse cells or primary oocytes in *D. virilis*. These possibilities could be distinguished via genetic ablation or knockdown of *Cid5* in females.

What is the molecular basis of the specialization of Cid paralogs in *D. virilis*? Although Cid1 and Cid5 have diverged in their HFDs, we speculate that the primary mode of specialization is via the much greater divergence of their N-terminal tail domains (NTDs). Cid1 and Cid5 differ significantly in both the retention of ancestral conserved motifs and in their acquisition of new motifs in their NTDs (Fig S6). All single copy *Cid* genes in *Drosophila* (including from *D. melanogaster*) encode a highly stereotyped set of protein sequence motifs 1–4 in their NTDs (Kursel & Malik, 2017; Malik et al, 2002), which are also conserved in *D. virilis* Cid1. Motifs 1–3 have been implicated in sister centromere cohesion in male meiosis (Collins et al, 2018) whereas motif 4 has been associated with BubR1 recruitment (Torras-Llort et al, 2010). However, Cid5 proteins have lost motifs 1 and 3 (Kursel & Malik, 2017), suggesting that these motifs are not necessary for Cid5's role in male meiosis and spermiogenesis. Differences in Cid1 and Cid5's NTDs could result in different protein interactions and cell type–specific kinetochore formation. Furthermore, both Cid1 and Cid5 have gained new motifs (Fig S6, motif 8 in Cid1, motifs 9 and 10 in Cid 5) not found in any Cid proteins encoded by single copy genes (e.g., Cid in *D. melanogaster*) (Kursel & Malik, 2017). We speculate that these "new" motifs may represent "degron-like" domains that underlie the specific loss of Cid1 and Cid5 in male and female meiosis, respectively, in *D. virilis*.

It is also possible that specialization of Cid1 and Cid5 could be a result of changes in the HFD. Although the HFD faces significant evolutionary constraints as a structural component of the nucleosome, some regions of the HFD are less well conserved. One such region is the loop (loop 1) between the first and second α helix of the HFD. In *D. melanogaster*, the loop 1 region of Cid has been shown to mediate interactions with the Cid chaperone, Cal1 (Chen et al, 2015; Rosin & Mellone, 2016). Interestingly, Cid1 and Cid5 differ at 7 of 14 loop 1 amino acids (Fig S6). However, Cal1 has not undergone duplication in *D. virilis*. Thus, it is likely that the same chaperone helps deposit both Cid paralogs in *D. virilis*. A third method of specialization could be through binding of distinct subcentromeric or peri-centromeric DNA sequences. Although Cid1 and Cid5 do broadly co-localize, their fluorescent signals are not perfectly coincident (e.g., Fig 2D). Additional experiments such as extended chromatin fibers and super-resolution imaging or ChIP-seq with Cid1 and Cid5 antibodies from germline tissues will shed light on this possibility.

Our findings may also reveal how centromere-drive may manifest deleterious consequences in *Drosophila* species. In previous work, we showed that *D. virilis* Cid1 but not *Cid5* evolves under positive selection (Kursel & Malik, 2019 *Preprint*). If *Cid1*-positive selection were driven by suppression of the deleterious effects of centromere drive, this would suggest the exciting possibility that the primary cost of centromere drive may not be manifest in male meiosis but instead in another life-stage. Indeed, work from *Arabidopsis* showed that distantly related CenH3 orthologs were capable of supporting mitosis and meiosis in *A. thaliana* CenH3 null plants. However, embryonic defects emerged when embryos were generated from pollen and ovules carrying different CenH3 orthologs (Maheshwari et al, 2015). In this scenario, Cid5 could relieve the evolutionary pressures on Cid1 to maintain the essential function of sperm centromeric identity.

Our study of Cid paralogs in *D. virilis* suggests that Cid1 and Cid5 carry out distinct roles in different cell types. Given the disparate nature of the chromatin environment in which each paralog functions, we propose that Cid1 and Cid5 face different selective pressures. Moreover, we propose that single copy *CenH3* genes must encode all of the functions performed by Cid1 and Cid5. If these roles are equally important for fitness, a single *CenH3* gene encoding both functions could become "trapped" for suboptimal function in both roles, for example, soma versus sperm. Such "intralocus conflict" occurs in the case of sexual genetic conflicts, whereby a locus beneficial in one sex is detrimental to the other (VanKuren & Long, 2018). However, the same functional tradeoff might also result if a single gene had two functional optima that could not be simultaneously achievable. One way to resolve intralocus conflict is through gene duplication and specialization of different paralogs for different functions (Gallach & Betran, 2011). For example, genes encoding mitochondrial function may have divergent optima in the soma versus testis, and this divergence has been invoked to explain the high retention rate of testis-specific gene paralogs encoding mitochondrial function (Gallach et al, 2010). If CenH3 function is also subject to dual constraints, then the duplication and specialization of different Cid paralogs in species like *D. virilis* may represent a more optimal state than the single copy *Cid* gene in species like *D. melanogaster*. Under this

scenario, the Cid1 and Cid5 paralogs of *D. virilis* provide an elegant example of nature's "separation-of-function" experiment for a CenH3 gene that has multiple essential functions in multicellular organisms.

# Materials and Methods

### Cid1 and Cid5 antibody production

We raised an antibody against Cid1 residues 15–31 (KSESHLDN-VEDSYEKTA) and Cid5 residues 56–71 (NLESPVAGEEPAPDTV). These sites were selected because they are in regions where Cid1 and Cid5 share no apparent homology and are distinct from other *D. virilis* proteins. Covance Inc. immunized two rabbits with the conjugated Cid5 peptide by injecting it four times over the course of 4 mo. Covance also immunized two rabbits for the Cid1 peptide by injecting it five times over the course of 5 mo. Our previous analysis of *D. virilis* Cid5 polymorphism revealed nonsynonymous variation in the Cid5 peptide sequence used to generate the antibody, therefore, for all experiments we ensured that we used *D. virilis* strains and cell lines have appropriate Cid5 alleles.

### Western blots from *D. virilis* WR DV-1 cells

*D. virilis* WR DV-1 cells were collected in RIPA (radioimmunoprecipitation assay) buffer and sonicated. Protein was quantified by Bradford assay and 20 μg total protein was analyzed by Western blot. We probed the membrane with either rabbit anti-Cid1 (1:2,000), rabbit anti-Cid5 (1:2,000), or rabbit anti-H3 (1:5,000 ab1791; Abcam) primary antibodies followed by goat anti-rabbit IgG-HRP (1:5,000; Santa Cruz Biotechnologies Inc.).

### Antibody staining of *D. virilis* tissue culture cells

We confirmed that the Cid1 antibody works for cytology by immunostaining *D. virilis* WR DV-1 cells. Cells were transferred to coverslips and fixed in 4% PFA for 5 min and blocked with PBSTx (0.3% Triton) plus 3% BSA for 30 min at room temperature. Then cells were incubated with primary antibodies at 4°C overnight. Coverslips with cells were incubated with secondary antibodies for 1 h at room temperature. Antibodies were diluted as follows: rabbit anti-Cid1 1:5,000 and (Alexa Fluor 568 A-11011; Invitrogen) 1:2,000.

### *D. virilis* transgenics

Cid1GFP and Cid5mCherry were cloned into a vector backbone containing piggyBac inverted repeats and the miniwhite gene cassette. This vector was generated by first removing 3XP3EGFP from the nosGal4-MW-pBacns plasmid (stock number 1290; Drosophila Genomics Resources Center). The 3XP3EGFP was removed as follows: nosGal4-MW-pBacns was digested with AgeI and AsiSI, run on a gel, and the largest band was gel isolated. Overhangs were blunted, and then the vector was ligated to itself to produce nosGal4_MWonly. Then the nanosGal4 cassette was removed as follows: nosGal4_MWonly was digested with NotI, run on a gel and

the largest band was gel isolated. Then the vector was ligated to itself to produce NoPromoter_miniwhite. Cid1GFP or Cid5mCherry, each with ~1 kb sequence upstream and downstream, were inserted between the AvrII and SbfI sites of the NoPromoter_miniwhite plasmid. For both Cid1 and Cid5, fluorescent proteins were inserted immediately five-prime of the RRRK motif at the beginning of the HFD. Fluorophores were flanked on both sides by three glycine residues to function as flexible linkers. Cid1GFP and Cid5mCherry plasmids were injected along with the piggyBac helper plasmid phspBac (Handler & Harrell, 1999) into *D. virilis* embryos. Injected flies were screened for red eye color. Injections and screening were performed by Rainbow Transgenics.

We attempted to knockout and knockdown Cid5 using CRISPR and miRNA-mediated approaches, respectively. For CRISPR knockout, we attempted to replace the Cid5 coding sequence with 3XP3dsRED. The injection mix contained Cas9 protein, a plasmid-based homology template and in vitro transcribed gRNAs. We failed to recover any progeny from injected animals carrying the fluorescent eye marker. For miRNA-mediated knockdown, we generated inducible Cid5-specific miRNAs under the control of the UAS-Gal4 expression system. We obtained stable *D. virilis* lines containing miRNAs using piggyBac mediated integration. We attempted to induce expression of the miRNAs using *D. virilis* Nanos-Gal4 (Drosophila Species Stock Number 15010-1051.102). However, we found weak and variable expression of Gal4 by the Nanos driver and were unable to obtain robust induction of the miRNA. A recent study found similarly unreliable expression of Cas9 using Nanos in the closely related *Drosophila novamexicana* (Lamb et al, 2020).

### Cytology

#### *General data collection and presentation practices*

For all cytological data, we present representative images acquired from the Leica TCS SP5 II confocal microscope with LASAF software and present maximally projected image files. For protein localization in larval neuroblasts, ovaries, testes, and the early embryo, a minimum of five organs and five cell types were examined for each assay of each stage.

### Preparation of larval neuroblasts for imaging and immunofluorescence

To assess Cid1 and Cid5 localization in larval brains, we used both Cid1- and Cid5-specific antibodies and Cid1GFP and Cid5mCherry transgenes. Brains from actively crawling third-instar larvae were dissected in PBS and transferred to 0.5% sodium citrate hypotonic solution 10 min. We transferred brains to a drop chromosome isolation buffer (120 mg MgCl$_2$:6H$_2$O, 1 g citric acid, 1 ml Triton X-100, and distilled H$_2$O to 100 ml) on a glass slide and fragmented the brains with needles for 4 min. Next, we lowered a coverslip was lowered onto the fragmented brains and squashed the brains under gentle pressure for 30 s. We then froze slides in liquid nitrogen. Then, slides were removed from liquid nitrogen and the coverslip was flipped off with a razor blade. Slides were immediately immersed in cold methanol for 5 min, cold acetone for 1 min and PBS for 1 min at room temperature. For experiments obtaining fluorescent signal from transgenes only, we removed the PBS and

added mounting medium with DAPI. For antibody staining, after incubation in acetone, brains were rinsed once in PBS and then incubated in PBS + 1% Triton X for 10 min for permeabilization. Slides were blocked in PBS + 0.1% Triton X + 3% BSA for 30 min at room temperature. Slides were incubated with primary antibody overnight at 4°C. Then slides were washed and incubated with secondary antibodies for 1 h at room temperature. Antibodies were diluted as follows: rabbit anti-Cid1 1:1,000, rabbit anti-Cid5 1:1,000, and Alexa Fluor goat anti-rabbit 568 1:1,000.

## Preparation of testes for imaging and immunofluorescence

To assess Cid1 and Cid5 localization in testes, we used Cid1 and Cid5-specific antibodies or transgenic flies encoding Cid1GFP or Cid5mCherry (both with internal tags and expressed under the control of their native promoters, as described above). To characterize Cid1 and Cid5 localization without antibody staining, we dissected testes in PBS from sexually mature (~10-d old) Cid1GFP, Cid5mCherry, or Cid1GFP/Cid5mCherry males. Testes were spread out on charged microscope slide, squashed under a coverslip and immediately immersed in liquid nitrogen. Testes were then fixed in 4% PFA for 7 min or cold methanol (5 min) and acetone (5 min). Testes were then mounted in SlowFade Gold antifade with DAPI. For immunofluorescence, we fixed testes from Cid1GFP or Cid5mCherry transgenic flies in 4% PFA. Testes were permeabilized in PBS + 0.3% Triton X for 30 min (2–15 min washes) and blocked in PBS + 0.1% Triton X + 3% BSA for 30 min at room temperature. Primary antibodies were diluted in block and incubated with testes overnight. Secondary antibodies were incubated in block for 1 h at room temperature. Antibodies were diluted as follows: mouse anti-phospho-histone H3 serine 10 (1:1,000 clone 3H10; Millipore) and Alexa Fluor goat anti-mouse 633 (1:1,000).

## Preparation of ovaries for imaging and immunofluorescence

To assess Cid1 and Cid5 localization in ovaries, we used Cid1 and Cid5-specific antibodies or transgenic flies encoding Cid1GFP or Cid5mCherry (as described above). To characterize Cid1 and Cid5 localization without antibody staining, we dissected ovaries in PBS from sexually mature (~10-d old) Cid1GFP, Cid5mCherry, or Cid1GFP/Cid5mCherry *D. virilis* females. Ovaries were fixed in 1:1 paraPBT:heptane (paraPBT = 4% PFA in PBS + 0.1% TritonX) for 10 min at room temperature. Then ovaries were washed, including one wash with 1X DAPI and mounted in SlowFade Gold (Thermo Fisher Scientific). For immunofluorescence, we performed fixation as above. We then blocked ovaries in PBS + 0.1% Triton X + 3% BSA for 30 min at room temperature. Ovaries were incubated with primary antibodies overnight at 4°C. Ovaries were then washed and incubated with secondary antibodies for 1 h at room temperature. Then ovaries were washed and mounted as above. Antibody dilutions were as follows: rabbit anti-Cid1 1:1,000, rabbit anti-Cid5 1:1,000, and Alexa Fluor goat anti-rabbit 568 1:1,000.

## Embryo collection, fixation, immunofluorescence, and imaging

To characterize Cid1 and Cid5 in the early embryo we imaged embryos produced from mothers and fathers with both Cid1GFP

and Cid5mCherry transgenes. 0–60-min-old embryos were collected on grape agar plates. Embryos were incubated in 30% bleach for 2 min to remove chorion. Fixation and antibody staining was performed according to Fanti and Pimpinelli method 3 (Fanti & Pimpinelli, 2004). Briefly, embryos were transferred to a 1:1 mixture of heptane and methanol and shaken vigorously for 1 min. The heptane layer was removed and embryos were washed twice with ice-cold methanol. Embryos rehydrated in PBS plus a drop of PBS + 0.1% Triton. Next, embryos were permeablized in PBS + 1% Triton for 30 min at room temperature. Embryos were blocked PBS + 0.1% Triton X + 3% BSA (BSA block) for 1 h at room temperature. We diluted primary antibodies in BSA block and incubated overnight at 4°C. Embryos were washed and then incubated with secondary antibodies diluted in BSA block for 2 h at room temperature. We washed embryos again after incubation with secondary antibodies and mounted embryos in wash solution (PBST). Embryos were imaged immediately after mounting. Primary antibody dilutions were the following: rabbit anti-AcH4 (1:1,000; Millipore), Alexa-Fluor goat secondary antibodies (Life Technologies) were diluted at 1:1,000.

# Supplementary Information

# Acknowledgements

We are grateful to Barbara Wakimoto for valuable discussions and advice. We would like to thank Ines Drinnenberg, Michelle Hays, Rini Kasinathan, Mia Levine, Antoine Molaro, Courtney Schroeder, Janet Young, and three anonymous reviewers for their comments on the manuscript. We thank the *Drosophila* Genetics Resource Center for the *D. virilis* tissue culture cells, Rainbow Transgenic Flies Inc. for the *D. virilis* transgenic injections and Covance Inc. for generating the polyclonal antibodies and the National *Drosophila* Species Stock Center (San Diego/Cornell) for the *D. virilis* flies. We also thank Yikang Rong and Justin Blumenstiel for the kind gift of the *Hiphop-GFP D. virilis* strains. This work was supported by funding from the National Institutes of Health training grants T32 HG000035 and T32 GM007270 (to LE Kursel) and R01 GM074108 (to HS Malik). The funders played no role in study design, data collection and interpretation, or the decision to publish this study. HS Malik is an Investigator of the Howard Hughes Medical Institute. The authors declare that they have no conflict of interest.

## Author Contributions

LE Kursel: conceptualization, data curation, formal analysis, supervision, funding acquisition, validation, investigation, visualization, methodology, project administration, and writing—original draft, review, and editing.
H McConnell: formal analysis, validation, investigation, methodology, and writing—review and editing.
AFA de la Cruz: formal analysis, supervision, investigation, visualization, methodology, and writing—review and editing.
HS Malik: conceptualization, supervision, funding acquisition, validation, visualization, project administration, and writing—original draft, review, and editing.

## Conflict of Interest Statement

The authors declare that they have no conflict of interest.

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
