## [Reviewer comments · Life Science Alliance]

Life Science Alliance

Gametic specialization of centromeric histone paralogs in *Drosophila virilis*

Lisa Kursel, Hannah McConnell, Aida de la Cruz, and Harmit Malik

DOI: <https://doi.org/10.26508/lsa.202000992>

Corresponding author(s): Harmit Malik, Fred Hutchinson Cancer Research Center/HHMI

Review Timeline:

Submission Date:	2020-12-14
Editorial Decision:	2020-12-15
Revision Received:	2021-04-03
Editorial Decision:	2021-04-21
Revision Received:	2021-04-27
Accepted:	2021-04-28

Scientific Editor: Shachi Bhatt

Transaction Report:

Please note that the manuscript was reviewed at Review Commons and these reports were taken into account in the decision-making process at *Life Science Alliance*.

Review
COMMONS

We are grateful to the editors and all three reviewers for their generous comments and suggestions. In response, we have compiled this Revision plan for your consideration; all proposed revisions that we undertake to make are highlighted with underlining.

Reviewer #1

(Evidence, reproducibility and clarity (Required)):

The manuscript by Kursel et al describes the tissue-specific localization of Cid1 and Cid5, two CENP-A paralogs, in *Drosophila virilis*. Using antibodies and fluorescence-tagged transgenes, the authors show that Cid1 is found in somatic cells and germ lines, whereas expression of Cid5 is restricted to germ lines. Moreover, female gametes retain only Cid1, while male gametes only retain Cid5. The latter rapidly disappears after fertilization, and only Cid1 is detectable in the zygote. The authors propose that duplication of an ancestral Cid allowed the emergence of tissue-specific expression, and potentially function, as a response to different constraints in the male and female germ lines.

The manuscript is very well written and clearly structured, and the data is nicely presented and of high quality, and to a very large part support the conclusions. I only have a few suggestions, mainly to improve clarity.

We are grateful for the positive comments of the reviewer and for the helpful suggestions to improve the revision.

****Major points:****

1. The authors state in the abstract and in the text (lines 19, 266, 302, 314, 325, 761, 802) that Cid1 replaces Cid5 after fertilization. The phrasing suggests that Cid1 takes the place of Cid5 at centromeres, but this is not actually shown. The authors cannot detect Cid5 anymore even in the very early zygote, where Cid1 is already present on the paternal chromatin. This is a detail, but it is relevant for the epigenetic inheritance model the authors propose, and the statements should therefore be more carefully phrased or more clearly explained.

We agree with this comment by the reviewer. We do not actually know whether Cid1 takes the place of Cid5 after fertilization because we cannot detect Cid5 after fertilization, only on mature sperm. Our model is based on previous work in *D. melanogaster* (Raychaudhuri *et al.* 2012), which showed that a significant depletion of Cid in sperm leads to defective centromere specification on paternal centromeres post-fertilization. Since we do not detect Cid1 in mature sperm, we therefore propose that Cid5 in sperm may carry out an analogous role of specifying paternal centromeres. To address this comment, we propose to amend our discussion and clarify that our interpretation is a hypothesis, which synthesizes our and previous findings.

2. The authors seem to assume that Cid1 and Cid5 localize to the same genomic regions, presumably the centromere. However, in some of the nuclei where both proteins are present, the signal is not completely overlapping, e.g. in Fig 3C. Do the authors think that these differences are real, or are these preparation artifacts?

Cid1 and Cid5 co-localize at the sub-micron-scale in our confocal z-stacks and they are the only two centromeric histones. However, the reviewer is correct that at this cytological resolution, we can't exclude the possibility that they localize to distinct sub-centromeric, or even pericentric regions. We also agree that the Cid1 and Cid5 signal is not perfectly co-incident in Fig 3C. It's possible that this is a staining artifact, but we can't rule out the possibility of underlying biological differences. To address this, we propose to revise the text to acknowledge the possibility that Cid1 and Cid5 might localize to distinct, centromeric regions. This possibility does not affect any of the other conclusions we present.

3. Based on the proposed role of Cid5 in epigenetic inheritance through the male germ line, the presence of the protein in nurse cells and oocytes in primary arrest in the female germ line (Fig 2C,D)

is surprising. Do the authors have an explanation for its function in these cells? It is an interesting observation that should be revisited in the discussion section.

We currently do not have an explanation for Cid5 presence in the female germline. Indeed, based on the RNA-seq and RT-PCR results, we were surprised to find such a robust pattern. It is possible that Cid5 presence is an evolutionary relic of its gene duplication from Cid1. Thus, Cid5 diverged from its ancestral, ubiquitous expression pattern to adopt a more germline-specific (both male and female) expression pattern. Alternatively, Cid5 could perform some function in nurse cells and primary oocytes; we cannot currently test this possibility due to lack of a genetic knockout. We propose to add these alternatives to a revised discussion section as suggested by the reviewer.

4. Lines 320-322. "Our study reveals that *D. virilis* employs a dedicated CenH3 paralog, Cid5, specifically for the purpose of epigenetic inheritance of centromere identity through the protamine-rich sperm chromatin environment." This statement goes too far, since the authors do not show that Cid5 carries epigenetic information. This should be rephrased to reflect the findings of the current study.

This comment is similar to point 1 above. We agree with the reviewer's comment and we propose to revise the text to indicate that this is a hypothesis and is based in part on previous work in *D. melanogaster*.

5. Lines 369-370. "Our study of Cid paralogs in *D. virilis* reveals that Cid1 and Cid5 carry out distinct roles in different cell types." In absence of functional analyses, the authors cannot claim that they reveal different roles for Cid1 and Cid5.

We agree. We propose to revise this statement to indicate that Cid1 and Cid5 may carry out centromeric functions in distinct cell types.

6. The authors state that they tried and failed to carry out knockdown experiments to analyze the function of the proteins. It would be helpful if they could elaborate why these experiments failed. Would it be possible to use the *D. virilis* culture cells to test if expression of Cid5 can rescue loss of Cid1? Alternatively, the authors could use available *Drosophila melanogaster* cell culture systems to at least test whether DvCid1 and/or DvCid5 rescue the loss of DmCid and are both competent in centromere function. This would also allow them to test if eventual differences in function are driven by the differences in the N-terminal tails.

The reviewer makes several helpful suggestions here, which we address individually.

We tried both a knockout and knockdown approach to assess the function of Cid5; both experiments failed (we believe) for technical reasons. Both experiments required extensive tool engineering as very few genetic reagents exist in the non-model *D. virilis*. For CRISPR knockout, our strategy was to replace the Cid5 coding sequence with a fluorescent protein that would be expressed in the eye. This approach relied on homology directed repair of a fairly large (>2kb) genomic fragment. We screened progeny from hundreds of injected G0s, but found no transmission of the fluorescent marker. We suspect that either cutting or repair was very inefficient. Moreover, in the absence of a means to track any knockout allele, follow-up cytological and genetic studies would be rendered impossible. For knockdown experiments, we generated inducible Cid1 and Cid5 specific miRNAs under the control of the UAS-Gal4 expression system. In collaboration with Rainbow Transgenics, we were able to obtain stable *D. virilis* lines containing both miRNAs using piggyBac mediated integration. To express the Cid1 or Cid5 miRNAs we relied on a pre-existing *D. virilis* line, Nanos-Gal4, which expresses Gal4 under a germline specific promoter. Unfortunately, we found weak and variable expression of Gal4 by the Nanos driver. A recent published study by Dr. Patricia Wittkopp (Univ. Michigan) found similarly unreliable expression of Cas9 using Nanos in the closely related *D. novamexicana* (Lamb *et al.* Front. Ecol. Evol. 2020). As a result, we were not able to obtain robust expression of the miRNAs. Knockdown in *D. virilis* and related species will likely require development of more robust driver lines for inducible expression. We propose to add a brief description of our knockout and knockdown efforts in the methods section in the revised manuscript and indicate that they failed for technical reasons. Like the Wittkopp paper, our description will provide a cautionary note to these published reagents in *D. virilis*.

Regarding tissue culture experiments: testing whether Cid5 can rescue Cid1 knockdown in *D. virilis* tissue culture cells is in theory a good experiment. However, we believe that it is outside the scope of a reasonably timed revision due to technical limitations. While we have been able to reliably transfect *D. virilis* cells (Kursel

and Malik 2017), these transfections are inefficient (~<1% of cells are transfected). A knockdown and replacement experiment would require a much higher transfection efficiency.

Finally, 'swap' experiments in *D. melanogaster* have been done previously (Vermaak *et al.* Mol. Cell. Biol. 2002, Rosin and Mellone Dev Cell 2016). These studies found that only closely related Cid orthologs are capable of localizing to centromeres in *D. melanogaster* tissue culture cells. *D. virilis* Cid cannot localize to *D. melanogaster* centromeres. Rosin and Mellone showed that this failure of localization was due to an incompatibility with the Cid chaperone, Cal1. Therefore, due to the incompatibility with *D. melanogaster* Cal1, experiments in *D. melanogaster* cells that test the function or localization of *D. virilis* Cid1 and Cid5 are unlikely to work.

****Minor points:****

7. Line 75. "Organisms in which CenH3 has duplicated and may have partitioned..." There may be words missing from this sentence.

We will edit this sentence to make it clearer that organisms with specialized Cid proteins may present an opportunity to study the tissue specific functions of Cid.

8. Line 297. "Only" twice in the sentence.

We will correct this in the revision.

9. Fig. 1E. Does the image show a metaphase spread or a metaphase nucleus?

This is not a metaphase spread. It is likely two distinct metaphase nuclei. We imaged cells from *D. virilis* larval brains, which have a high number of dividing cells, and identified cells with condensed chromosomes. We will update the figure legend to clarify that these are metaphase nuclei.

10. Fig. S4C. Twice "male" in the crossing scheme.

Thank you for pointing out this error. We will correct this in the revision.

Reviewer #1 (Significance (Required)):

Despite their conserved and essential function, centromeres and centromeric proteins are very diverse, and the description of additional animal model systems is therefore of importance for the understanding of centromere evolution. The identification of Cid proteins with male and female gamete-restricted localization is very interesting, and supports the model that the authors and others have previously proposed for the specialization of CENP-A paralogs. However, the absence of functional analyses makes the conclusions about the roles of these proteins rather speculative, and limits the scope of the findings.

****Referees cross-commenting****

My review (Reviewer 1) raised points that are partially overlapping with those of the other reviewers, and I agree with all of their comments.

The results presented here form the basis for follow-up work to understand how centromeres adapt to meiotic constraints and are inherited through the germ lines. The manuscript will therefore be of great interest for researchers studying epigenetic inheritance, centromere drive and genetic conflicts between male and female meiosis.

We appreciate the comments of the reviewer. Ours is the first report of a female versus male gamete-restricted localization of a centromeric protein (at the limit of cytological detection). We acknowledge that genetic knockouts/knockdowns would enable a more robust understanding of the specialization but that is not currently technically feasible in this species.

Reviewer #2

(Evidence, reproducibility and clarity (Required)):

****Summary:****

This manuscript investigates the functional specialisation of two centromeric histone (Cid) paralogs in *Drosophila virilis*. Based on previous observations in plants, and RNA expression studies in flies, the authors put forward the hypothesis that Cid paralogs have acquired germline specific functions in this animal. They use cytological tools (specific antibodies and FP-tagged transgenic lines) to investigate Cid1 and Cid5 localisation in male and female germlines. They convincingly show that oocyte centromeres contain Cid1 (and not Cid5) and mature sperm centromeres contain Cid5 (and not Cid1). The authors go on to follow the transgenerational inheritance of Cid5, to determine if it marks centromere identity on paternal chromosomes. They propose that in the zygote, Cid5 on the male pronucleus would be gradually replaced by maternally supplied Cid1. Surprisingly, they find that this is not the case. Instead, Cid5 is not detectable on the male pronucleus, whereas Cid1 is immediately visible, which differs from previous results in *Drosophila melanogaster*.

We thank the reviewer for their generous comments on the gametic specialization of the Cid paralogs and the accurate summary of the article.

****Major comments:****

I have two major comments on this study. The first is related to the precise cell cycle timing of Cid1/Cid5 removal in male/female germlines. The second is related to the timing of Cid1 localisation in the male pronucleus post fertilisation.

1. The authors propose metaphase of meiosis I to be the critical transition point at which Cid5 is removed from oocytes and Cid1 is removed from spermatocytes. However, improved temporal resolution of this removal event is required.

In oocytes, images showing that Cid5 is lost and that Cid1 is retained by stage 14 are convincing. But is not clear whether the removal occurs at or prior to metaphase of meiosis I i.e. during prophase or prometaphase I. Moreover, is Cid5 removal gradual or dramatic i.e. stage specific? In Figure 2D, which stage oocyte is shown?

In males, Cid1 appears to be lost by metaphase of meiosis I. Again, it would be interesting from a mechanistic perspective to know whether this loss is gradual during prophase to prometaphase I or rapid at metaphase I. In the specific case of the transgene, Cid1GFP foci are visible at metaphase of meiosis I, but are lost at the leaf stage (Figure 3, S3B). For a complete picture, the authors should show whether Cid1 is absent or present at centromeres in interphase or during meiosis II. If Cid1 is absent at meiosis II then this would indicate that Cid5 is capable of supporting cell division, which might indicate more about Cid5 function. Apart from PH3S10P, are there any available cytological markers in *D. virilis* to confirm that cells are in prometaphase/metaphase of meiosis I or II?

Finally, it would be interesting to know whether Cid5 and Cid1 localise in the earliest germ cell precursors i.e. germline stem cells. Presumably this is the first point at which Cid5 is expressed and localised to centromeres in development.

The reviewer raises an important point that we have not yet addressed in the manuscript: a more precise staging of the Cid5 removal during oogenesis and Cid1 removal during spermiogenesis.

Regarding the oocyte: Figure 2D, in which the oocyte is in mid-prophase I (stage 5 – based on egg chamber morphology), shows both Cid1 and Cid5 present on the oocyte nucleus. We also know that by stage 14, Cid5 is undetectable. We propose to address the question of exactly when Cid5 is removed by using the Cid5 antibody and transgene to examine oocyte stages between stage 5 and stage 14. These results will allow us to address at what stage Cid5 is removed and whether Cid5 removal is gradual or stage-specific. We will also update figures and figure legends to clarify the precise stages shown.

Regarding Cid1 removal in the male germline: we propose to use cytological tools to look more closely at meiosis I and meiosis II in the male germline. In addition to the PH3S10 antibody, we can use a tubulin antibody to look for meiotic spindles. This should give us a clear indication of MI vs MII. We will determine

whether Cid1 is present in pro-metaphase I, metaphase I, prometaphase II and metaphase II and also whether its removal is gradual or dramatic.

To determine where Cid5 is first expressed, we propose to perform IF experiments targeting the male and female germline stem cells. To aid in the identification of the germline stem cells we will stain Cid5-mCherry ovaries and testes with anti-spectrin. The germline stem cells should be identifiable by the presence of a 'spectrosome', a spherical cytoskeletal organelle. In the male germline, we will also try to co-stain with Fasciculin III, which illuminates the Hub (the somatic cell neighbors of the GSCs) cells in *D. melanogaster*. Although we have not tested either antibody in *D. virilis*, the high degree of amino acid identity between *D. melanogaster* and *D. virilis* spectrin and Fasciculin III make them potentially good candidates to investigate.

2. The finding that Cid5 is not detectable on the male pronucleus is unexpected and suggests that it is rapidly removed upon fertilisation. Is it possible to track this Cid5 'removal' at even earlier time points post fertilisation?

For example, do the dynamics of Cid5 removal correlate with the timing of either H3 (H3.3) incorporation during protamine to histone exchange or the first round of DNA replication prior to the first mitosis? Also is the loading of Cid1 on the male pronucleus gradual rather than rapid? It appears so if you compare Figure 5B and 5C.

The authors claim that Cid levels are equalised between male and female pronuclei at apposition. This does appear to be the case, but it would help to highlight male and female Cid1 foci in Figure 5E.

Related to Figure 5F, do the authors have any evidence that new Cid1 loading occurs between anaphase and early G1 phase as might be expected at this cell cycle time? The foci do appear more intense.

We appreciate the reviewer's suggestion to more closely track the timing of Cid5 removal in the early embryo. However, the earliest developmental stages of the embryo are challenging to track. Since early development happens extremely rapidly, the embryos must be collected immediately after fertilization. We have dedicated considerable effort to getting *D. virilis* flies to rapidly mate and lay eggs. Our embryo collection experiments already capture the earliest possible window after fertilization since eggs were collected within 30 minutes of laying. Given the technical limits to collecting embryos any sooner, we can only speculate the Cid5 is removed at a stage earlier than we can visualize, since it is present on mature sperm but not in fertilized zygotes. In the revision, we propose to add data that tracks major remodeling events of the paternal genome including H3.3 incorporation and paternal genome replication. We used an Ach4 antibody to help identify the paternal pronucleus in our embryo experiments. This allowed us to see that, even at the earliest stages we collected (for example, Figure 5B), the paternal genome had already transitioned to a nucleosome-based chromatin organization. Therefore, we speculate that Cid5 is removed either before or concurrent with H3.3-H4 loading.

Regarding Cid1 dynamics on the male pronucleus and in the early embryo in general: we agree with the reviewer's suggestion that initial Cid1 loading on the paternal genome appears to be gradual. We propose to more explicitly point this out in the revised manuscript. We also agree that our data is consistent with loading of Cid1 during embryonic anaphase as has been previously shown for Cid in *D. melanogaster* (Schuh *et al.* 2007). We will add text and appropriate references to the revised manuscript putting our data in the context of what is known about Cid dynamics in the *D. melanogaster* early embryo.

In addition, as the reviewer suggests, we will highlight male and female Cid1 foci in Figure 5E.

Finally, given that the authors fail to detect Cid5 on the paternal genome, is it possible that some Cid1 remains at mature sperm centromeres at an undetectable level? In this case, Cid1 would specify centromere identity and Cid5 function might be dispensable post-fertilisation and rather be required to maintain centromeres in the context of a protamine-rich environment.

The reviewer is right that it is technically possible that a cytologically undetectable amount of Cid1 remains on the paternal genome. However, in *D. melanogaster*, when Cid is diluted below cytological detection, it is unable to maintain centromere identity on paternal genomes. Therefore, in the absence of Cid1, we propose that Cid5

must still be required to carry out these functions. We will add the caveat about cytological detection in our revision.

****Minor comments:****

3. The experiment showing the over-expression and localisation of Venus-Cid5 in *D. mel* Kc cells is not particularly helpful or well-explained. The inclusion of anti-Cid5 antibody staining in Figure S3 would confirm that the antibody is working and specifically recognises Cid5 and not Cid1 for example at the leaf stage. If possible, it would be useful if the authors included an additional centromere marker to confirm Cid1 or Cid5 localisation to centromeres e.g. Figure S3, Figure 3D, Figure 5.

We thank the reviewer for pointing out the confusing antibody validation experiment. We agree that a better test of the Cid5 antibody would be to show its localization to centromeres *in vivo* in *D. virilis*. In the revised manuscript, we will clarify the antibody validation and include better descriptions of this validation *in vivo*. We will update Figure S3 to include Cid5 antibody staining; showing the stages of sperm development with Cid1GFP and Cid5 antibody at each stage.

We are somewhat limited by the fact that there are not many *D. melanogaster* antibodies that work as additional centromere markers in *D. virilis*. Both *D. virilis* CENP-C and Cal1 are highly diverged from their *D. melanogaster* orthologs; as a result, antibodies raised to *D. melanogaster* CENP-C do not work in *D. virilis*. Our best centromere-adjacent marker is HipHop-GFP, which we do use of in our manuscript already. In the revised manuscript, we will add images of MI-metaphase from HipHopGFP, Cid5mCherry males, similar to what is in Figure 3D. We expect that the Cid5 signal will co-localize with one of the two HipHopGFP foci per chromosome as it does in sperm.

4. In the discussion, perhaps the authors could comment in the structure/function of the centromere assembly factors CAL1 in *D. virilus*. Are L1 loops of the HFD of both Cid1 and Cid5 predicted to bind CAL1?

To our knowledge, there is no way to predict whether a specific loop 1 sequence will interact with Cal1. Cid1 and Cid5 only share three amino acids in common in the Loop 1 region of their histone fold domains. However, while both Cid and CENP-C have duplicated in *D. virilis*, Cal1 has not, suggesting it may act on both Cid paralogs. In the revised manuscript, we will include discussion of how Cal1 might interact with multiple Cid paralogs.

5. In the discussion, the authors mention that knockdown or knockouts experiments were unsuccessful. But it was unclear whether this was specific to Cid paralogs or that these techniques are not efficient in this species. The authors should clarify this.

The knockdown and knockout experiments failed for technical reasons, as we described in detail above in our response to Reviewer 1 – Point 6. We propose to clarify this and add a description of the knockdown strategy in the revised Methods section.

Reviewer #2 (Significance (Required)):

****Nature of advance:****

In most organisms, the centromeric histone CENP-A specifies centromere identity and function. Most organisms encode one CENP-A gene. However, some organisms which have duplicated CENP-A, for example plants, show evidence for specialisation of paralogs, particularly in the germline. CENP-A has different functional requirements and dynamics in the germ line. For example, in oocytes arrested in meiosis or on mature sperm that has undergone protamine exchange. Although CENP-A paralogs are rare in animals, this manuscript shows that *D. virilis* harbours two Cid paralogs with differential localisation patterns on mature male and female gametes. This study opens up the possibility for germline specification of CENP-A paralogs in other animals. It also raises questions as to how centromeric histones might be actively removed from centromeres, indicating that meiosis I might be a critical cell cycle transition that might allow such a reorganisation. It is striking that the transgenerational inheritance of functional centromeres appears to differ between *D. melanogaster* and *D. virilis*. This opens up the possibility that different mechanisms might operate in other animals.

We are grateful to the reviewer for providing the right context to discuss our findings.

****Audience:**** Those interested in genetics, inheritance, chromosome biology, epigenetics and evolution.

****Expertise:**** Centromere assembly and maintenance in germ cells (germline stem cells and meiosis).

****Referees cross-commenting****

My review (Reviewer 2) is in agreement with those of the two other reviewers. In particular, I agree with Reviewer 1 concerning the statement that *cid1* replaces *cid5* after fertilisation. This needs to be rephrased as this has not definitively been shown, but is important for understanding the epigenetic specification of centromeres in the male germline.

We agree with both reviewers and we will rephrase as requested.

Reviewer #3

(Evidence, reproducibility and clarity (Required)):

Summary: In this manuscript, Kursel et al. investigate the specialization of two Cid paralogues, Cid1 and Cid5 in *D. virilis*. They look at Cid1 and Cid5 localization in different tissues: somatic cells, testis, ovaries and early embryo. Their analyses reveal that while in somatic cells only Cid1 is expressed, there is a gametic specialization. Only Cid1 is retained in mature female gamete, whereas only Cid5 is retained in mature male gametes. In addition, they show the rapid replacement of Cid5 by Cid1 following fertilization. These results support the conclusion that there is gametic specialization of Cid paralogs in *D. virilis*.

The manuscript is clear and well-written. The results are robust: they used two different approaches (antibodies and transgenic lines) to confirm their observations. We only have minor comments.

We are grateful to the reviewer for their generous appraisal and comments.

****Minor comments:****

- Line 56: please define the abbreviation before using it (MI).

We will update the manuscript to say "Meiosis I (MI)" to introduce the abbreviation.

- Line 68: It would be nice to have more background about centromere drive hypothesis, it's a fascinating but complex concept.

We will provide additional background on the centromere drive hypothesis in the introduction.

- Line 176 : refer to Figure S2H not 2H.

We will correct this error; thank you.

- In Supplemental figure 2 panel G and H : typo error, missing "a" in metaphase.

We will correct this typo; thank you.

- In Supplemental Figure 4C : typo error , crossing two males together.

We will correct this error.

- Line 335-337 : It is possible that, while Cid5 is specialized to function in male germline, it may not have completely lost its ancestral function. Cid1 may be able to replace Cid5 in the event of Cid5 knockdown. Cid1 is indeed expressed earlier. One could imagine a competition between Cid1 and Cid5 based on, for example, chromatin environment.

We cannot rule out the possibility that Cid1 could compensate for loss of Cid5. However, Cid5 has lost motifs in its N-terminal tail that are otherwise completely conserved among *Drosophila* Cid1 proteins suggesting it is unlikely that Cid5 could complement loss of Cid1 (Kurse and Malik MBE 2017). Moreover, Cid1 and Cid5 do appear to coexist in earlier stages of both male and female gametogenesis. Thus, it is not the presence of Cid5 but some other stage-dependent factor (possibly chromatin environment) that is leading to the 'active removal'

of Cid1 in the male germline. Thus, we believe that it's unlikely that Cid1 would persist throughout spermatogenesis even if we were to eliminate Cid5.

- Line 353-358: It would be helpful to the reader to include a little more information about the function of the NTD vs HFD make it clear why the NTD is more likely to be related to the specialization. In addition, a schematic of the two proteins with their different domains may help to visualize the variation.

We thank the reviewer for this suggestion. In the revised manuscript, we will add several sentences elaborating on why we think specialization would most likely occur via the N-terminal tail. The main reason is that Cid1 and Cid5 share almost no sequence identity in their N-terminal tails. They also each encode a unique set of conserved motifs, which we described previously (Kursel and Malik MBE 2017). We will also include Cid1 and Cid5 protein schematics show their unique motifs and that indicate percent amino acid similarity in the N-terminal tail vs. the Histone Fold Domain.

- The authors could mention in the discussion (probably in the last paragraph) that young duplicated genes tend to have testes-biased expression (Assis and Bachtrog 2013), which could favor gametic specialization.

This point is related to Reviewer 1 - Point 3. In the revised manuscript, we will add a paragraph to the discussion section that speculates on what Cid5s expression pattern might have been immediately after duplication. As Reviewer 3 mentions, young genes tend to have testis-biased expression patterns. If this was the case for Cid5, it suggests that Cid5 subsequently acquired female germline expression. Another possibility is that Cid5 was born with general germline expression (male and female). This latter possibility provides one explanation for why Cid5 is expressed in nurse cells and female oocyte in prometaphase. We will add the suggested reference (Assis and Bachtrog 2013), to this new discussion paragraph.

- In the Discussion there are two points which are not addressed but would be interesting to provide some perspective on:

- First, the gamete specialization occurs only in late gametogenesis. Both Cid1 and Cid5 are expressed before metaphase I, in male and female. How do you interpret this? Can this give you some clues about the underlying molecular mechanisms? You mention this in the results but we think it's a very interesting point which deserves to be discussed in more detail in the Discussion.

We thank the reviewer for their suggestion to include two additional interesting discussion points. First, how do we interpret the presence of both Cid paralogs during the first half of male and female gametogenesis. One explanation, as we discussed above, is that Cid5 was born with a germline (both male and female) promoter. Perhaps, as specialization occurred, presence of Cid5 was selected against late in oogenesis and presence of Cid1 was selected against during the second half of spermiogenesis. This scenario suggests that the presence of both paralogs in early stages of gametogenesis is a relic of Cid5s original expression pattern. Alternatively, it's possible that Cid1 and Cid5 both perform important functions early in gametogenesis. One possibility is that Cid1 acts as a template for Cid5 deposition in males if Cid5 is not competent for centromere establishment. We will add these Discussion points in our revision.

- Second, we are surprised that the authors didn't mention the centromere drive hypothesis in their discussion. It would be interesting to interpret these results in light this hypothesis. Is it possible that gametic specialization of Cid can prevent the male fitness cost of centromere drive? But paradoxically, would it not favor the emergence of centromere drive in absence of suppression? It would be very interesting to have the authors' point of view here.

We agree with the reviewer that adding some points related to centromere-drive would be interesting. In previous work (Kursel and Malik MBE 2017), we showed that Cid1 but not Cid5 evolves under positive selection. Thus, if Cid1 positive selection were driven by it acting as a suppressor of centromere drive, then this would suggest that the presence of Cid5 could relieve the pressures on Cid1 to maintain the function of sperm centromeric identity. The reviewer is also right that this would further suggest the exciting possibility that the primary cost of centromere drive may not be manifest in male meiosis but instead in another life-stage. Indeed, work from *Arabidopsis* showed that distantly related CenH3 orthologs were capable of supporting mitosis and meiosis in *A. thaliana* CenH3 null plants. However, embryonic defects emerged when embryos were generated from pollen and ovules carrying different CenH3 orthologs (Maheshwari, PLoS Genetics,

2015). Therefore, it's possible that driving centromeres would still be selected against in *D. virilis* if the costs of drive manifest in early embryonic division. We will add these discussion points to the revised manuscript.

Reviewer #3 (Significance (Required)):

This paper adds more evidence for a role of gene duplication in resolving intra-locus conflicts. These authors have previously shown that Cid had independently duplicated in some *Drosophila* and mosquito species. In these papers, they hypothesized that Cid duplication allows for tissue or cell-type-specific expression to resolve a conflict. In a previous paper on Cid paralogs in *Drosophila virilis*, they showed evidence that Cid5 and Cid1 were expressed differently across tissues using RT-PCR.

Here the authors test their hypothesis using new tools that allow them to precisely localize transcripts and analyze the differences between Cid paralogs in *Drosophila virilis*. This precision allowed them to map expression to cell types within the ovaries and testes and pinpoint, for example, meiosis I as an important transition point between putative functions of Cid1 and Cid5. This is a necessary advance towards understanding the functions and evolution of paralogs in this important gene family. Among the more exciting results is the replacement of paternally-inherited Cid5 with Cid1 very early on in embryos-right after fertilization and before the first mitotic division. The key evidence for the different roles of the Cid paralogs would be having different phenotypes on knockout or knockdown, but even in the absence of these experiments (the authors were not able to make a Cid5 KD), their results are exciting.

This system offers a promising model to study the specificity of chromosome segregation in mitosis vs meiosis. In that respect, we think this paper would interest a broad audience of evolutionary and cellular/developmental biologists.

We thank the reviewer for their generous comments.

Expertise: *Drosophila* molecular and evolutionary genetics

****Referees cross-commenting****

There is some overlap between my review (Reviewer 3) and the other reviews, and I agree with each of their comments. Both Reviewers 1 and 2 raise important points about the precise timing of Cid1 and Cid5 removal that the authors should carefully consider. I also think that Reviewer 2 makes a great point about the possibility that some Cid1 remains at mature sperm centromeres but is not detectable.

We agree with these points and have addressed them above.

December 15, 2020

Re: Life Science Alliance manuscript #LSA-2020-00992-T

Dr. Harmit S Malik
Fred Hutchinson Cancer Research Center/HHMI
Division of Basic Sciences
Division of Basic Sciences Fred Hutchinson Cancer Research Center 1100 Fairview Avenue N. A1-162
A2-025
Seattle, WA 98109

Dear Dr. Malik,

Thank you for submitting your manuscript entitled "Gametic specialization of centromeric histone paralogs in *Drosophila virilis*" to Life Science Alliance (LSA).

For a brief overview: this manuscript was reviewed via Review Commons, and then transferred to LSA. After assessing the reviewers' enthusiastic comments and the authors' point-by-point response to the reviewers' comments, the manuscript was deemed appropriate to be published in LSA, once the authors have addressed the reviewers concerns in accordance to their revision plan laid out in the pbp response. We invite you to submit a revised manuscript that addresses the reviewers' points.

Thank you for this interesting contribution to Life Science Alliance. We are looking forward to receiving your revised manuscript.

Sincerely,

Shachi Bhatt, Ph.D.
Executive Editor
Life Science Alliance
<https://www.lsjournal.org/>
Tweet @SciBhatt @LSAJournal

- A letter addressing the reviewers' comments point by point.
- An editable version of the final text (.DOC or .DOCX) is needed for copyediting (no PDFs).
- High-resolution figure, supplementary figure and video files uploaded as individual files: See our detailed guidelines for preparing your production-ready images, <https://www.life-science-alliance.org/authors>
- Summary blurb (enter in submission system): A short text summarizing in a single sentence the study (max. 200 characters including spaces). This text is used in conjunction with the titles of papers, hence should be informative and complementary to the title and running title. It should describe the context and significance of the findings for a general readership; it should be written in the present tense and refer to the work in the third person. Author names should not be mentioned.

B. MANUSCRIPT ORGANIZATION AND FORMATTING:

Editor

After assessing the reviewers' enthusiastic comments and the authors' point-by-point response to the reviewers' comments, the manuscript was deemed appropriate to be published in LSA, once the authors have addressed the reviewers concerns in accordance to their revision plan laid out in the pbp response.

We are grateful to the editors (and all three reviewers) for their generous comments and suggestions. We have added experiments and revised the writing in the few sections as suggested by the reviewers.

Reviewer #1

(Evidence, reproducibility and clarity (Required)):

The manuscript by Kursel et al describes the tissue-specific localization of Cid1 and Cid5, two CENP-A paralogs, in *Drosophila virilis*. Using antibodies and fluorescence-tagged transgenes, the authors show that Cid1 is found in somatic cells and germ lines, whereas expression of Cid5 is restricted to germ lines. Moreover, female gametes retain only Cid1, while male gametes only retain Cid5. The latter rapidly disappears after fertilization, and only Cid1 is detectable in the zygote. The authors propose that duplication of an ancestral Cid allowed the emergence of tissue-specific expression, and potentially function, as a response to different constraints in the male and female germ lines.

The manuscript is very well written and clearly structured, and the data is nicely presented and of high quality, and to a very large part support the conclusions. I only have a few suggestions, mainly to improve clarity.

We are grateful for the positive comments of the reviewer and for the helpful suggestions to improve the revision.

****Major points:****

1. The authors state in the abstract and in the text (lines 19, 266, 302, 314, 325, 761, 802) that Cid1 replaces Cid5 after fertilization. The phrasing suggests that Cid1 takes the place of Cid5 at centromeres, but this is not actually shown. The authors cannot detect Cid5 anymore even in the very early zygote, where Cid1 is already present on the paternal chromatin. This is a detail, but it is relevant for the epigenetic inheritance model the authors propose, and the statements should therefore be more carefully phrased or more clearly explained.

We agree with this comment by the reviewer. We do not actually know whether Cid1 takes the place of Cid5 after fertilization because we cannot detect Cid5 after fertilization, only on mature sperm. Our model is based on previous work in *D. melanogaster* (Raychaudhuri *et al.* 2012), which showed that a significant depletion of Cid in sperm leads to defective centromere specification on paternal centromeres post-fertilization. Since we do not detect Cid1 in mature sperm, we therefore propose that Cid5 in sperm may carry out an analogous role of specifying paternal centromeres. We have revised our manuscript (including all lines indicated by the reviewer) to clarify these points. Specifically, we no longer say that Cid1 replaces Cid5, rather we say that Cid5 is the only detectable Cid protein on *D. virilis* sperm, and that Cid1 is rapidly loaded onto paternal centromeres post fertilization. This leads us to hypothesize that Cid5 is involved in transgenerational inheritance.

2. The authors seem to assume that Cid1 and Cid5 localize to the same genomic regions, presumably the centromere. However, in some of the nuclei where both proteins are

present, the signal is not completely overlapping, e.g. in Fig 3C. Do the authors think that these differences are real, or are these preparation artifacts?

Cid1 and Cid5 co-localize at the sub-micron-scale in our confocal z-stacks and they are the only two centromeric histones. However, the reviewer is correct that at this cytological resolution, we can't exclude the possibility that they localize to distinct sub-centromeric, or even pericentric regions. In the discussion section of our revised manuscript, we now acknowledge the possibility that Cid1 and Cid5 might localize to distinct, centromeric regions. This possibility does not affect any of the other conclusions we present.

3. Based on the proposed role of Cid5 in epigenetic inheritance through the male germ line, the presence of the protein in nurse cells and oocytes in primary arrest in the female germ line (Fig 2C,D) is surprising. Do the authors have an explanation for its function in these cells? It is an interesting observation that should be revisited in the discussion section.

We currently do not have an explanation for Cid5 presence in the female germline. Indeed, based on the RNA-seq and RT-PCR results, we were surprised to find such a robust pattern. It is possible that Cid5 presence is an evolutionary relic of its gene duplication from Cid1. Thus, Cid5 diverged from its ancestral, ubiquitous expression pattern to adopt a germline-specific (both male and female) expression pattern. Alternatively, Cid5 could perform some function in nurse cells and primary oocytes; we cannot currently test this possibility due to lack of a genetic knockout. Have now added a revised discussion section highlighting these points.

4. Lines 320-322. "Our study reveals that *D. virilis* employs a dedicated CenH3 paralog, Cid5, specifically for the purpose of epigenetic inheritance of centromere identity through the protamine-rich sperm chromatin environment." This statement goes too far, since the authors do not show that Cid5 carries epigenetic information. This should be rephrased to reflect the findings of the current study.

This comment is similar to point 1 above. We agree with the reviewer's comment and have revised the text to indicate that this is a hypothesis based on previous work from *D. melanogaster*.

First paragraph of discussion:

*Our study suggests that *D. virilis* employs a dedicated CenH3 paralog, Cid5, specifically for the purpose of epigenetic inheritance of centromere identity through the protamine-rich sperm chromatin environment. This specialization was accomplished following gene duplication by gain of germline-specific expression of Cid5 and specific removal of the ancestral Cid1 during the prometaphase to metaphase transition of MI in the male germline. It is technically possible that a small amount of Cid1 remains in mature sperm beyond our cytological limit of detection via multiple methods. However, degradation of *D. melanogaster* Cid in the male germline below the limits of cytological detection results in early embryonic lethality (Raychaudhuri et al 2012). Thus, undetectably low levels of Cid1 are unlikely to be functionally sufficient for epigenetic inheritance of centromere identity. Therefore, we hypothesize that Cid5 must provide the epigenetic mark of centromeres across generations. Although Cid1 no longer appears to perform centromere inheritance function through the male germline in *D. virilis*, it still serves CenH3 function in both the soma and the female germline.*

5. Lines 369-370. "Our study of Cid paralogs in *D. virilis* reveals that Cid1 and Cid5 carry out distinct roles in different cell types." In absence of functional analyses, the authors cannot claim that they reveal different roles for Cid1 and Cid5.

We agree. We have revised the text to indicate that it is possible that Cid1 and Cid5 carry out distinct roles. These lines now read: “*Our study of Cid paralogs in D. virilis raises the possibility that Cid1 and Cid5 carry out distinct roles in different cell types*”

6. The authors state that they tried and failed to carry out knockdown experiments to analyze the function of the proteins. It would be helpful if they could elaborate why these experiments failed. Would it be possible to use the *D. virilis* culture cells to test if expression of Cid5 can rescue loss of Cid1? Alternatively, the authors could use available *Drosophila melanogaster* cell culture systems to at least test whether DvCid1 and/or DvCid5 rescue the loss of DmCid and are both competent in centromere function. This would also allow them to test if eventual differences in function are driven by the differences in the N-terminal tails.

The reviewer makes several helpful suggestions here, which we address individually.

We tried both knockout and knockdown approach to assess the function of Cid5; both experiments failed (we believe) for technical reasons. We now include a brief description of our knockout and knockdown efforts in the methods section in the revised manuscript and indicate that they failed for technical reasons.

The text added to the methods section is as follows:

We attempted to knockout and knockdown Cid5 using CRISPR and miRNA mediated approaches, respectively. For CRISPR knockout, we attempted to replace the Cid5 coding sequence with 3XP3dsRED. The injection mix contained Cas9 protein, a plasmid-based homology template and in vitro transcribed gRNAs. We failed to recover any progeny from injected animals carrying the fluorescent eye marker. For miRNA-mediated knockdown, we generated inducible Cid5 specific miRNAs under the control of the UAS-Gal4 expression system. We obtained stable D. virilis lines containing miRNAs using piggyBac mediated integration. We attempted to induce expression of the miRNAs using D. virilis Nanos-Gal4 (Drosophila Species Stock Number 15010-1051.102). However, we found weak and variable expression of Gal4 by the Nanos driver and were unable to obtain robust induction of the miRNA. A recent found similarly unreliable expression of Cas9 using Nanos in the closely related D. novamexicana (Lamb et al. Front. Ecol. Evol. 2020)

Regarding tissue culture experiments: testing whether Cid5 can rescue Cid1 knockdown in *D. virilis* tissue culture cells is in theory a good experiment. However, we believe that it is outside the scope of a reasonably timed revision due to technical limitations. While we have been able to reliably transfect *D. virilis* cells (Kursel and Malik 2017), these transfections are inefficient (~<1% of cells are transfected). A knockdown and replacement experiment would require a much higher transfection efficiency.

Finally, ‘swap’ experiments in *D. melanogaster* have been done previously (Vermaak et al. Mol. Cell. Biol. 2002, Rosin and Mellone Dev Cell 2016). These studies found that only closely related Cid orthologs are capable of localizing to centromeres in *D. melanogaster* tissue culture cells. *D. virilis* Cid cannot localize to *D. melanogaster* centromeres. Rosin and Mellone showed that this failure of localization was due to an incompatibility with the Cid chaperone, Cal1. Therefore, due to the incompatibility with *D. melanogaster* Cal1, experiments in *D. melanogaster* cells that test the function or localization of *D. virilis* Cid1 and Cid5 are unlikely to work.

****Minor points:****

7. Line 75. "Organisms in which CenH3 has duplicated and may have partitioned..." There may be words missing from this sentence.

We have edited this sentence to make it clearer that organisms with specialized Cid proteins may present an opportunity to study the tissue specific functions of Cid.

8. Line 297. "Only" twice in the sentence.

We have corrected this_in the revision.

9. Fig. 1E. Does the image show a metaphase spread or a metaphase nucleus?

This is not a metaphase spread. It is likely two distinct metaphase nuclei. We imaged cells from *D. virilis* larval brains, which have a high number of dividing cells, and identified cells with condensed chromosomes. We have updated the figure legend to clarify that these are metaphase nuclei.

10. Fig. S4C. Twice "male" in the crossing scheme.

Thank you for pointing out this error. We have corrected the crossing scheme (now Fig. S5C) in the revision.

Reviewer #1 (Significance (Required)):

Despite their conserved and essential function, centromeres and centromeric proteins are very diverse, and the description of additional animal model systems is therefore of importance for the understanding of centromere evolution. The identification of Cid proteins with male and female gamete-restricted localization is very interesting, and supports the model that the authors and others have previously proposed for the specialization of CENP-A paralogs. However, the absence of functional analyses makes the conclusions about the roles of these proteins rather speculative, and limits the scope of the findings.

****Referees cross-commenting****

My review (Reviewer 1) raised points that are partially overlapping with those of the other reviewers, and I agree with all of their comments.

The results presented here form the basis for follow-up work to understand how centromeres adapt to meiotic constraints and are inherited through the germ lines. The manuscript will therefore be of great interest for researchers studying epigenetic inheritance, centromere drive and genetic conflicts between male and female meiosis. We appreciate the comments of the reviewer. Ours is the first report of a female versus male gamete-restricted localization of a centromeric protein (at the limit of cytological detection). We acknowledge that genetic knockouts/knockdowns would enable a more robust understanding of the specialization but that is not currently feasible.

Reviewer #2

(Evidence, reproducibility and clarity (Required)):

****Summary:****

This manuscript investigates the functional specialisation of two centromeric histone (Cid) paralogs in *Drosophila virilis*. Based on previous observations in plants, and RNA expression studies in flies, the authors put forward the hypothesis that Cid paralogs

have acquired germline specific functions in this animal. They use cytological tools (specific antibodies and FP-tagged transgenic lines) to investigate Cid1 and Cid5 localisation in male and female germlines. They convincingly show that oocyte centromeres contain Cid1 (and not Cid5) and mature sperm centromeres contain Cid5 (and not Cid1). The authors go on to follow the transgenerational inheritance of Cid5, to determine if it marks centromere identity on paternal chromosomes. They propose that in the zygote, Cid5 on the male pronucleus would be gradually replaced by maternally supplied Cid1. Surprisingly, they find that this is not the case. Instead, Cid5 is not detectable on the male pronucleus, whereas Cid1 is immediately visible, which differs from previous results in *Drosophila melanogaster*.

We thank the reviewer for their generous comments on the gametic specialization of the Cid paralogs and the accurate summary of the article.

****Major comments:****

I have two major comments on this study. The first is related to the precise cell cycle timing of Cid1/Cid5 removal in male/female germlines. The second is related to the timing of Cid1 localisation in the male pronucleus post fertilisation.

The reviewer raises several important points that we address in order.

1. The authors propose metaphase of meiosis I to be the critical transition point at which Cid5 is removed from oocytes and Cid1 is removed from spermatocytes. However, improved temporal resolution of this removal event is required.

In oocytes, images showing that Cid5 is lost and that Cid1 is retained by stage 14 are convincing. But is not clear whether the removal occurs at or prior to metaphase of meiosis I i.e. during prophase or prometaphase I. Moreover, is Cid5 removal gradual or dramatic i.e. stage specific? In Figure 2D, which stage oocyte is shown?

To address this in our revised manuscript we examined oocyte stages between stage 5 and stage 14. We found that we could routinely detect Cid5mCherry through stage 12 (Figure S2C – S2E). We were also able to detect Cid5mCherry in two out of six Stage 14 oocyte nuclei (Figure S2F). We ranked each of the 6 stage 14 nuclei from early stage 14 to late stage 14 by measuring the distance across the long axis of the oocyte nucleus (assuming that later stage 14 will be more elongated as the oocyte nucleus enters MI – metaphase arrest). We found that the two oocyte nuclei that contained Cid5mCherry were among the earliest stage 14 nuclei (Figure S2F, S2G). These results suggest the Cid5 removal in the ovary is stage-specific and occurs during the MI – metaphase. We have updated the text and added several figure panels to reflect these new findings (Figure S2). We have also updated figures and figure legends to clarify the precise stages shown in each panel.

In males, Cid1 appears to be lost by metaphase of meiosis I. Again, it would be interesting from a mechanistic perspective to know whether this loss is gradual during prophase to prometaphase I or rapid at metaphase I. In the specific case of the transgene, Cid1GFP foci are visible at metaphase of meiosis I, but are lost at the leaf stage (Figure 3, S3B). For a complete picture, the authors should show whether Cid1 is absent or present at centromeres in interphase or during meiosis II. If Cid1 is absent at meiosis II then this would indicate that Cid5 is capable of supporting cell division, which might indicate more about Cid5 function. Apart from PH3S10P, are there any available cytological markers in *D. virilis* to confirm that cells are in prometaphase/metaphase of meiosis I or II?

To address this in our revised manuscript we looked more closely at the transition from MI – prophase to MI – metaphase. We found that Cid1 is detectable in the latest stages of MI – prophase but absent from MI – metaphase (using PH3S10 and Cid1 antibodies). We have added a panel to Figure 3 documenting this transition (Figure 3F). We conclude that, like Cid5 removal in the female germline, Cid1 removal in the male germline is stage-specific and occurs upon entry into MI – metaphase. We have updated the text and figures with these additional findings.

Finally, it would be interesting to know whether Cid5 and Cid1 localise in the earliest germ cell precursors i.e. germline stem cells. Presumably this is the first point at which Cid5 is expressed and localised to centromeres in development.

To answer this question in our revision, we used anti-Spectrin to identify GSCs in the male and female germline (based on the presence of the spherical 'spectrosome' organelle). We found that Cid5 is first detectable in the GSCs in males and females. Notably, we also found that Cid5 is absent from the somatic Hub cells in males and the somatic Terminal Filament and Cap cells in females. We have added several figure panels highlighting this. Figure 2A shows that Cid1GFP but not Cid5mCherry is present in the somatic Terminal Filament and Cap cells at the apical tip of the germarium (based on nuclear morphology). Cid5mCherry is detectable in this image but only in the neighboring cells. Figure 2C identifies these neighboring cells as the female GSCs based on anti-spectrin staining and shows Cid5mCherry staining in the GSCs. Similarly, Figure 3C shows a group of cells at the apical tip of the testis where Cid1 is detectable but Cid5mCherry is not. Figure 3D shows that these cells are the somatic Hub cells and that Cid5 first appears in the neighboring GSCs which we also identify using anti-Spectrin.

2. The finding that Cid5 is not detectable on the male pronucleus is unexpected and suggests that it is rapidly removed upon fertilisation. Is it possible to track this Cid5 'removal' at even earlier time points post fertilisation?

We appreciate the reviewer's suggestion to more closely track the timing of Cid5 removal in the early embryo. However, the earliest developmental stages of the embryo are challenging to track. Since early development happens extremely rapidly, the embryos must be collected immediately after fertilization. We have dedicated considerable effort to getting *D. virilis* flies to rapidly mate and lay eggs. Our embryo collection experiments already capture the earliest possible window after fertilization since eggs were collected within 30 minutes of laying. Given the technical limits to collecting embryos any sooner, we can only speculate the Cid5 is removed at a stage earlier than we can visualize, since it is present on mature sperm but not in fertilized zygotes.

For example, do the dynamics of Cid5 removal correlate with the timing of either H3 (H3.3) incorporation during protamine to histone exchange or the first round of DNA replication prior to the first mitosis?

In our revised manuscript, we have added text and a supplementary figure (Figure S4) that present our early embryo data in the context of major remodeling events of the paternal genome. We used an Ach4 antibody to identify the paternal pronucleus in our embryo experiments. This allowed us to see that, even at the earliest stages we collected (for example, Figure 5B), the paternal genome had already transitioned to a nucleosome-based chromatin organization. Therefore, we speculate that Cid5 is removed either before or concurrent with H3.3-H4 loading.

Also is the loading of Cid1 on the male pronucleus gradual rather than rapid? It appears so if you compare Figure 5B and 5C.

We agree with the reviewer's suggestion that initial Cid1 loading on the paternal genome appears to be gradual. We have edited the text to more explicitly point this out in the revised manuscript.

The authors claim that Cid levels are equalised between male and female pronuclei at apposition. This does appear to be the case, but it would help to highlight male and female Cid1 foci in Figure 5E.

We have highlighted male and female Cid1 foci in Figure 5E as suggested.

Related to Figure 5F, do the authors have any evidence that new Cid1 loading occurs between anaphase and early G1 phase as might be expected at this cell cycle time? The foci do appear more intense.

We agree that our data is consistent with loading of Cid1 during embryonic anaphase as has been previously shown for Cid in *D. melanogaster* (Schuh *et al.* 2007b). We have added text and appropriate references to the revised manuscript, putting our data in the context of what is known about Cid dynamics in the *D. melanogaster* early embryo.

Finally, given that the authors fail to detect Cid5 on the paternal genome, is it possible that some Cid1 remains at mature sperm centromeres at an undetectable level? In this case, Cid1 would specify centromere identity and Cid5 function might be dispensable post-fertilisation and rather be required to maintain centromeres in the context of a protamine-rich environment.

The reviewer is right that it is technically possible that a cytologically undetectable amount of Cid1 remains on the paternal genome. However, in *D. melanogaster*, when Cid is diluted below cytological detection, it is unable to maintain centromere identity on paternal genomes. Therefore, in the absence of Cid1, we propose that Cid5 must still be required to carry out these functions. We have added the caveat about cytological detection in our revision as well as an explanation of why we do not favor that hypothesis.

****Minor comments:****

3. The experiment showing the over-expression and localisation of Venus-Cid5 in *D. mel Kc* cells is not particularly helpful or well-explained. The inclusion of anti-Cid5 antibody staining in Figure S3 would confirm that the antibody is working and specifically recognises Cid5 and not Cid1 for example at the leaf stage. If possible, it would be useful if the authors included an additional centromere marker to confirm Cid1 or Cid5 localisation to centromeres e.g. Figure S3, Figure 3D, Figure 5.

We thank the reviewer for pointing out the confusing antibody validation experiment. We have removed the experiment in *D. melanogaster* tissue culture cells from Figure S1 and added panel showing Cid5 antibody staining of early canoe stage sperm cells from Cid1GFP testes as the reviewer suggested (Figure S3B).

We are somewhat limited by the fact that there are not many *D. melanogaster* antibodies that work as additional centromere markers in *D. virilis*. Both *D. virilis* CENP-C and Cal1 are highly diverged from their *D. melanogaster* orthologs; as a result, antibodies raised to *D. melanogaster* CENP-C do not work in *D. virilis*. Our best centromere-adjacent marker is HipHop-GFP, which we do use of in our manuscript already. In the revised manuscript, we introduce HipHop-GFP earlier and use it as an additional centromeric marker in the mitotic zone of the testis (Figure S3A). As expected, Cid1 antibody staining co-localizes with a subset to HipHop-GFP foci.

4. In the discussion, perhaps the authors could comment in the structure/function of the

centromere assembly factors CAL1 in *D. virilis*. Are L1 loops of the HFD of both Cid1 and Cid5 predicted to bind CAL1?

To our knowledge, there is no way to predict whether a specific loop 1 sequence will interact with Cal1. Cid1 and Cid5 only share seven out of fourteen amino acids in common in the Loop 1 region of their histone fold domains. However, while both Cid and CENP-C have duplicated in *D. virilis*, Cal1 has not, suggesting it may act on both Cid paralogs. In the revised manuscript, we have included a discussion of how Cal1 might interact with multiple Cid paralogs. We have also included a supplementary figure (Figure S5) with a schematic that highlights the differences between Cid1 and Cid5 in their N-terminal tail domains and HFDs.

5. In the discussion, the authors mention that knockdown or knockouts experiments were unsuccessful. But it was unclear whether this was specific to Cid paralogs or that these techniques are not efficient in this species. The authors should clarify this.

The knockdown and knockout experiments failed for technical reasons, as we described in detail above in our response to Reviewer 1 – Point 6. We have added a paragraph to the methods section describing our efforts.

Reviewer #2 (Significance (Required)):

****Nature of advance:****

In most organisms, the centromeric histone CENP-A specifies centromere identity and function. Most organisms encode one CENP-A gene. However, some organisms which have duplicated CENP-A, for example plants, show evidence for specialisation of paralogs, particularly in the germline. CENP-A has different functional requirements and dynamics in the germ line. For example, in oocytes arrested in meiosis or on mature sperm that has undergone protamine exchange. Although CENP-A paralogs are rare in animals, this manuscript shows that *D. virilis* harbours two Cid paralogs with differential localisation patterns on mature male and female gametes. This study opens up the possibility for germline specification of CENP-A paralogs in other animals. It also raises questions as to how centromeric histones might be actively removed from centromeres, indicating that meiosis I might be a critical cell cycle transition that might allow such a reorganisation. It is striking that the transgenerational inheritance of functional centromeres appears to differ between *D. melanogaster* and *D. virilis*. This opens up the possibility that different mechanisms might operate in other animals.

We are grateful to the reviewer for providing the right context to discuss our findings.

****Audience:**** Those interested in genetics, inheritance, chromosome biology, epigenetics and evolution.

****Expertise:**** Centromere assembly and maintenance in germ cells (germline stem cells and meiosis).

****Referees cross-commenting****

My review (Reviewer 2) is in agreement with those of the two other reviewers. In particular, I agree with Reviewer 1 concerning the statement that cid1 replaces cid5 after fertilisation. This needs to be rephrased as this has not definitively been shown, but is important for understanding the epigenetic specification of centromeres in the male germline.

We agree with both reviewers and we have rephrased as requested.

Reviewer #3

(Evidence, reproducibility and clarity (Required)):

Summary: In this manuscript, Kursel et al. investigate the specialization of two Cid paralogues, Cid1 and Cid5 in *D. virilis*. They look at Cid1 and Cid5 localization in different tissues: somatic cells, testis, ovaries and early embryo. Their analyses reveal that while in somatic cells only Cid1 is expressed, there is a gametic specialization. Only Cid1 is retained in mature female gamete, whereas only Cid5 is retained in mature male gametes. In addition, they show the rapid replacement of Cid5 by Cid1 following fertilization. These results support the conclusion that there is gametic specialization of Cid paralogs in *D. virilis*.

The manuscript is clear and well-written. The results are robust: they used two different approaches (antibodies and transgenic lines) to confirm their observations. We only have minor comments.

We are grateful to the reviewer for their generous appraisal and comments.

****Minor comments:****

- **Line 56: please define the abbreviation before using it (MI).**

We now introduce the abbreviation as "Meiosis I (MI)."

- **Line 68: It would be nice to have more background about centromere drive hypothesis, it's a fascinating but complex concept.**

We now provide additional background on the centromere drive hypothesis in the introduction.

- **Line 176 : refer to Figure S2H not 2H.**

We corrected this error; thank you.

- **In Supplemental figure 2 panel G and H : typo error, missing "a" in metaphase.**

We corrected this error; thank you.

- **In Supplemental Figure 4C : typo error , crossing two males together.**

We corrected this error; thank you.

- **Line 335-337 : It is possible that, while Cid5 is specialized to function in male germline, it may not have completely lost its ancestral function. Cid1 may be able to replace Cid5 in the event of Cid5 knockdown. Cid1 is indeed expressed earlier. One could imagine a competition between Cid1 and Cid5 based on, for example, chromatin environment.**

We cannot rule out the possibility that Cid1 could compensate for loss of Cid5. However, Cid5 has lost motifs in its N-terminal tail that are otherwise completely conserved among *Drosophila* Cid1 proteins suggesting it is unlikely that Cid5 could complement loss of Cid1 (Kursel and Malik MBE 2017). Moreover, Cid1 and Cid5 do appear to coexist in earlier stages of both male and female gametogenesis. Thus, it is not the presence of Cid5 but some other stage-dependent factor (possibly chromatin environment) that is leading to the 'active removal' of Cid1 in the male germline. Thus, we believe that it's unlikely that Cid1 would persist throughout spermatogenesis even if we were to eliminate Cid5. However, in our revised Discussion, we now formally acknowledge the possibility that Cid1 may be retained on sperm in the absence of Cid5.

- **Line 353-358: It would be helpful to the reader to include a little more information about**

the function of the NTD vs HFD make it clear why the NTD is more likely to be related to the specialization. In addition, a schematic of the two proteins with their different domains may help to visualize the variation.

We thank the reviewer for this suggestion. In the revised manuscript, we have added several sentences to the discussion elaborating on why we think specialization would most likely occur *via* the N-terminal tail. The main reason is that Cid1 and Cid5 share almost no sequence identity in their N-terminal tails. They also each encode a unique set of conserved motifs, which we described previously (Kursel and Malik MBE 2017). We have also added a discussion of the possibility of specialization via the loop 1 region of the HFD (Cid loop 1 interacts with the chaperone, Cal1 in *D. melanogaster*), and of specialization by binding different sub-centromeric regions. Finally, as the reviewer suggested, we have included Cid1 and Cid5 protein schematics showing their unique N-terminal motif (Figure S5). This figure also shows percent amino acid similarity in the N-terminal tail vs. the Histone Fold Domain (Figure S5A) and an alignment of the loop 1 region of the HFD (Figure S5B).

- The authors could mention in the discussion (probably in the last paragraph) that young duplicated genes tend to have testes-biased expression (Assis and Bachtrog 2013), which could favor gametic specialization.

This point is related to Reviewer 1 - Point 3. In the revised manuscript, we have added a paragraph to the discussion section that speculates on what Cid5's expression pattern might have been immediately after duplication. As Reviewer 3 mentions, young genes tend to have testis-biased expression patterns. If this was the case for Cid5, it suggests that Cid5 subsequently acquired female germline expression. Another possibility is that Cid5 was born with general germline expression (male and female). This latter possibility provides one explanation for why Cid5 is expressed in nurse cells and female oocyte in prometaphase. We have added the suggested reference (Assis and Bachtrog 2013), to this new discussion paragraph.

- In the Discussion there are two points which are not addressed but would be interesting to provide some perspective on:

- First, the gamete specialization occurs only in late gametogenesis. Both Cid1 and Cid5 are expressed before metaphase I, in male and female. How do you interpret this? Can this give you some clues about the underlying molecular mechanisms? You mention this in the results but we think it's a very interesting point which deserves to be discussed in more detail in the Discussion.

To address this point, we now discuss the possibility that "Cid5's expression may have been driven by a GSC and germ-cell specific promoter that cannot distinguish between male and female germlines." Alternatively, both paralogs may perform some important function during early oogenesis and spermatogenesis, as we discuss. We also point out that the specialization depends not on the specific expression pattern but rather on the specific removal of Cid5 from oocytes and Cid1 from sperm; we hypothesize that this mechanism must be via some proteolytic degradation.

- Second, we are surprised that the authors didn't mention the centromere drive hypothesis in their discussion. It would be interesting to interpret these results in light this hypothesis. Is it possible that gametic specialization of Cid can prevent the male fitness cost of centromere drive? But paradoxically, would it not favor the emergence of centromere drive in absence of suppression? It would be very interesting to have the authors' point of view here.

We agree with this suggestion and have now added the following paragraph to our discussion:

Our findings may also reveal how centromere-drive may manifest deleterious consequences in Drosophila species. In previous work, we showed that D. virilis Cid1 but not Cid5 evolves under positive selection (Kursel & Malik 2019). If Cid1 positive selection were driven by suppression of the deleterious effects of centromere drive, this would suggest the exciting possibility that the primary cost of centromere drive may not be manifest in male meiosis but instead in another life-stage. Indeed, work from Arabidopsis showed that distantly related CenH3 orthologs were capable of supporting mitosis and meiosis in A. thaliana CenH3 null plants. However, embryonic defects emerged when embryos were generated from pollen and ovules carrying different CenH3 orthologs (Maheshwari et al 2015). In this scenario, Cid5 could relieve the evolutionary pressures on Cid1 to maintain the essential function of sperm centromeric identity.

Reviewer #3 (Significance (Required)):

This paper adds more evidence for a role of gene duplication in resolving intra-locus conflicts. These authors have previously shown that Cid had independently duplicated in some Drosophila and mosquito species. In these papers, they hypothesized that Cid duplication allows for tissue or cell-type-specific expression to resolve a conflict. In a previous paper on Cid paralogs in Drosophila virilis, they showed evidence that Cid5 and Cid1 were expressed differently across tissues using RT-PCR.

Here the authors test their hypothesis using new tools that allow them to precisely localize transcripts and analyze the differences between Cid paralogs in Drosophila virilis. This precision allowed them to map expression to cell types within the ovaries and testes and pinpoint, for example, meiosis I as an important transition point between putative functions of Cid1 and Cid5. This is a necessary advance towards understanding the functions and evolution of paralogs in this important gene family. Among the more exciting results is the replacement of paternally-inherited Cid5 with Cid1 very early on in embryos-right after fertilization and before the first mitotic division. The key evidence for the different roles of the Cid paralogs would be having different phenotypes on knockout or knockdown, but even in the absence of these experiments (the authors were not able to make a Cid5 KD), their results are exciting.

This system offers a promising model to study the specificity of chromosome segregation in mitosis vs meiosis. In that respect, we think this paper would interest a broad audience of evolutionary and cellular/developmental biologists.

We thank the reviewer for their generous comments.

Expertise: Drosophila molecular and evolutionary genetics

****Referees cross-commenting****

There is some overlap between my review (Reviewer 3) and the other reviews, and I agree with each of their comments. Both Reviewers 1 and 2 raise important points about the precise timing of Cid1 and Cid5 removal that the authors should carefully consider. I also think that Reviewer 2 makes a great point about the possibility that some Cid1 remains at mature sperm centromeres but is not detectable.

We agree with these points and have addressed them above.

April 21, 2021

RE: Life Science Alliance Manuscript #LSA-2020-00992-TR

Dr. Harmit S Malik
Fred Hutchinson Cancer Research Center/HHMI
Division of Basic Sciences
Division of Basic Sciences Fred Hutchinson Cancer Research Center 1100 Fairview Avenue N. A1-162
A2-025
Seattle, WA 98109

Dear Dr. Malik,

Thank you for submitting your revised manuscript entitled "Gametic specialization of centromeric histone paralogs in *Drosophila virilis*". We would be happy to publish your paper in Life Science Alliance pending final revisions necessary to meet our formatting guidelines.

Along with the points listed below, please also attend to the following:

- please add an Author Contributions section to your main manuscript text
- please add a conflict of interest statement to your main manuscript text
- please revise the legend for Figure 2, as there is no mention of panel F, which is provided in the actual figure
- please add callouts for Figures 5F; S2I and S4D to your main manuscript text
- please provide higher resolution better quality WBs for Fig 1A, CID5
- please add scale bars for Figure 4E and S2F

A. FINAL FILES:

-- High-resolution figure, supplementary figure and video files uploaded as individual files: See our detailed guidelines for preparing your production-ready images, <https://www.life-science->

alliance.org/authors

B. MANUSCRIPT ORGANIZATION AND FORMATTING:

Sincerely,

Shachi Bhatt, Ph.D.
Executive Editor
Life Science Alliance
<http://www.lsjournal.org>
Tweet @SciBhatt @LSAJournal

Reviewer #1 (Comments to the Authors (Required)):

1.Short Summary: This manuscript investigates the functional specialisation of two centromeric histone (Cid) paralogs in *Drosophila virilis*. Based on previous observations in plants, and RNA expression studies in flies, the authors put forward the hypothesis that Cid paralogs have acquired germline specific functions in this animal. They use cytological tools (specific antibodies and FP-tagged transgenic lines) to investigate Cid1 and Cid5 localisation in male and female germlines. They convincingly show that oocyte centromeres contain Cid1 (and not Cid5) and mature sperm centromeres contain Cid5 (and not Cid1). The authors go on to follow the transgenerational inheritance of Cid5, to determine if it marks centromere identity on paternal chromosomes. They propose that in the zygote, Cid5 on the male pronucleus would be gradually replaced by maternally supplied Cid1. Surprising, they find that this is not the case. Instead, Cid5 is not detectable on the male pronucleus, whereas Cid1 is immediately visible, which differs from previous results in *Drosophila melanogaster*.

Advance: This study opens up the possibility for germline specification of CENP-A paralogs in other animals. It also raises questions as to how centromeric histones might be actively removed from centromeres, indicating that meiosis I might be a critical cell cycle transition that might allow such a reorganisation. It is striking that the transgenerational inheritance of functional centromeres appears to differ between *D. melanogaster* and *D. virilis*. This opens up the possibility that different mechanisms might operate in other animals.

2. Comments on the revised version of the manuscript:

In this revised version, the authors have performed an excellent job at addressing the two major concerns I previously raised. This manuscript is now appropriate for publication in Life Science Alliance.

Firstly, the authors have provided a more precise and detailed temporal analysis of Cid5 localisation in females and Cid1 localisation in males. They can now define the removal point as metaphase of meiosis I in both sexes. This has added mechanistic insight into their observations. In addition, they also convincingly show that Cid5 is restricted to germ-lineage cells, first detectable in male and female GSCs.

Secondly, related to the timing of Cid1 localisation in the male pronucleus post fertilization, the authors have toned down their interpretation of results and have included of the caveat that paternal Cid1 might still be present at an undetectable level. The addition of the AcH4 staining of paternal nuclei also adds resolution to the timing of Cid5 removal and was satisfying to see.

Lastly, I also found the additional discussion of possible interactions with Cal1 particularly interesting and insightful.

3. There are no additional issues to be addressed.

April 28, 2021

RE: Life Science Alliance Manuscript #LSA-2020-00992-TRR

Dr. Harmit S Malik
Fred Hutchinson Cancer Research Center/HHMI
Division of Basic Sciences
Division of Basic Sciences Fred Hutchinson Cancer Research Center 1100 Fairview Avenue N. A1-162
A2-025
Seattle, WA 98109

Dear Dr. Malik,

Thank you for submitting your Research Article entitled "Gametic specialization of centromeric histone paralogs in *Drosophila virilis*". It is a pleasure to let you know that your manuscript is now accepted for publication in Life Science Alliance. Congratulations on this interesting work.

DISTRIBUTION OF MATERIALS:

Again, congratulations on a very nice paper. I hope you found the review process to be constructive and are pleased with how the manuscript was handled editorially. We look forward to future exciting submissions from your lab.

Sincerely,

Shachi Bhatt, Ph.D.

Executive Editor

Life Science Alliance

<http://www.lsjournal.org>
